# Spectral-Bias and Kernel-Task Alignment in Physically Informed Neural Networks

## Abstract

Physically informed neural networks (PINNs) are a promising emerging method for solving differential equations. As in many other deep learning approaches, the choice of PINN design and training protocol requires careful craftsmanship. Here, we suggest a comprehensive theoretical framework that sheds light on this important problem. Leveraging an equivalence between infinitely over-parameterized neural networks and Gaussian process regression (GPR), we derive an integro-differential equation that governs PINN prediction in the large data-set limit— the neurally-informed equation. This equation augments the original one by a kernel term reflecting architecture choices and allows quantifying implicit bias induced by the network via a spectral decomposition of the source term in the original differential equation.

## Introduction

Deep neural networks (DNNs) are revolutionizing a myriad of data-intensive disciplines (Krizhevsky et al., 2012; Jumper et al., 2021; Brown et al., 2020), offering optimization-based alternatives to handcrafted algorithms. The recent "physics-informed neural network" (PINN) approach (Raissi et al., 2019) trains DNNs to approximate solutions to partial differential equations (PDEs) by directly introducing the equation and boundary/initial conditions into the loss function, effectively enforcing the equation on a set of collocation points in the domain of interest. While still leaving much to be desired in terms of performance and robustness, its promise has already attracted broad attention across numerous scientific disciplines (Mao et al., 2020; Jin et al., 2021; Molnar et al., 2023; Patel et al., 2022; Coutinho et al., 2023; Ruggeri et al., 2022; Hamel et al., 2023; Cai et al., 2021; Karniadakis et al., 2021; Hao et al., 2022; Cai et al., 2021; Cuomo et al., 2022; Huang et al., 2022; Das & Tesfamariam, 2022).

Recent years have seen steady progress in our theoretical understanding of DNNs through various mappings to Gaussian Process Regression (GPR) (Hron et al., 2020; Naveh et al., 2021; Li & Sompolinsky, 2021; Naveh & Ringel, 2021; Seroussi et al., 2023; Hanin & Zlokapa, 2023), which are already carrying practical implications (Yang et al., 2021; Bahri et al., 2022; Maloney et al., 2022). For supervised learning tasks, including real-world ones, it was shown that kernel-task alignment or spectral bias (Cohen et al., 2021; Canatar et al., 2021) is strongly predictive of DNN performance. Specifically, diagonalizing the GPR kernel on the measure induced by the dataset leads to a series of features and eigenvalues. Those features with high eigenvalues can be learned with the least training effort.

An important line of work by Wang et al. (2021a; 2022; 2021b) generalized some of these findings. Specifically, an NTK-like (Jacot et al., 2018) correspondence with GPR was established. While this provided insights into spectral bias and architecture design (Wang et al., 2021b), the proposed formalism involves large dataset-dependent matrix inverses and does not easily lend itself to analytical predictions of generalization and robustness. Furthermore, these works do not provide general measures of kernel-task alignment.

In this paper, we provide a comprehensive theoretical framework for generalization in over-parametrized PINNs. This effort places the notions of spectral bias and kernel-task alignment in PINNs on solid theoretical footing and extends to a broad range of PDEs (including non-linear ones) and architectures. Our main result is the Neurally-Informed Equation (NIE), an integro-differential equation that approximates PINN predictions on test points. This equation fleshes out tangible links

between the DNN's kernel, the original differential equation, and PINN's prediction. In the limit of an infinite dataset, the neurally-informed equation reduces to the original equation, yet for large but finite datasets it captures the difference between the exact solution and the PINN prediction. It further offers several practical measures of kernel-task alignment which can provide robustness assurance for PINNs.

## 1 GENERAL SETTING

Consider the following well-posed partial differential equation (PDE) defined on a closed bounded domain, $\Omega \subseteq \mathbb{R}^d$

$$
\begin{aligned}
L[f](\boldsymbol{x}) &= \phi(\boldsymbol{x}) & \boldsymbol{x} \in \Omega \\
f(\boldsymbol{x}) &= g(\boldsymbol{x}) & \boldsymbol{x} \in \partial\Omega
\end{aligned}
\tag{1}
$$

where $L[f]$ is a differential operator and $f : \Omega \to \mathbb{R}$ is the PDE's unknown solution with $\boldsymbol{x} \in \mathbb{R}^d$. The functions $\phi : \Omega \to \mathbb{R}$ and $g : \partial\Omega \to \mathbb{R}$ are the PDE's respective source term and initial/boundary conditions. In what follows, we use bold font to denote vectors.

In the PINN setting, we choose a set of training points (collocation points) on the domain $\Omega$ and its boundary, $\partial\Omega$ around which we encourage a neural network to obey the differential equation. Specifically, we denote $X_n = \{\boldsymbol{x}_\mu\}_{\mu=1}^n$, where $n = n_{\partial\Omega} + n_\Omega$ and $n_\Omega, n_{\partial\Omega}$ respectively are the number of data points in the domain's bulk and on its boundary. We define the DNN estimator $\hat{f}_{\boldsymbol{\theta}} : \Omega \to \mathbb{R}$ of the solution $f$, where $\boldsymbol{\theta}$ are the network parameters. To train this DNN the following loss function is minimized,

$$
\mathcal{L} = \frac{1}{\sigma_\Omega^2} \sum_{\mu=1}^{n_\Omega} \left( L[\hat{f}_{\boldsymbol{\theta}}](\boldsymbol{x}_\mu) - \phi(\boldsymbol{x}_\mu) \right)^2 + \frac{1}{\sigma_{\partial\Omega}^2} \sum_{\nu=1}^{n_{\partial\Omega}} \left( \hat{f}_{\boldsymbol{\theta}}(\boldsymbol{x}_\nu) - g(\boldsymbol{x}_\nu) \right)^2,
\tag{2}
$$

where $\sigma_\Omega^2$, and $\sigma_{\partial\Omega}^2$ are some positive tunable constants. A good choice of hyper-parameters and network architecture should allow the network to minimize this training loss (i.e. respect the differential equation around the collocation points) and generalize to points outside the training set (i.e. obey the differential equation around an arbitrary point).

Obtaining analytical solutions to this optimization problem is obviously difficult for anything but the simplest linear neural networks. One analysis approach, which proved useful in the context of supervised learning, is to consider the limit of infinitely wide neural networks. Here, several simplifications arise, allowing one to map the problem to Bayesian Inference with GPR Jacot et al. (2018); Li & Sompolinsky (2021); Naveh et al. (2021); Mandt et al. (2017).

Specifically, we consider DNNs trained using *full-batch* gradient descent with weight decay and external white Gaussian noise. The discrete-time dynamics of the parameters are thus

$$
\boldsymbol{\theta}_{t+1} - \boldsymbol{\theta}_t = -\eta \left( \gamma \boldsymbol{\theta}_t + \nabla_{\boldsymbol{\theta}_t} \mathcal{L} \right) + 2\sigma\sqrt{\eta}\boldsymbol{\xi}_{t+1}
\tag{3}
$$

where $\boldsymbol{\theta}_t$ is the vector of all network parameters at time step $t$, $\gamma$ is the strength of the weight decay set so that $\sigma^2/\gamma$ is the desired initialization/prior weight variance, $\mathcal{L}$ is the loss as a function of the DNN parameters $\boldsymbol{\theta}_t$, and data , $\sigma$ is the magnitude of noise, $\eta$ is the learning rate and $\xi_{t,i}$ is a centred standardized Gaussian variable. As $\eta \to 0$ these discrete-time dynamics converge to the continuous-time Langevin equation given by $\dot{\boldsymbol{\theta}}(t) = -\nabla_{\boldsymbol{\theta}} \left( \frac{\gamma}{2}|\boldsymbol{\theta}(t)|^2 + \mathcal{L}(\boldsymbol{\theta}(t), \mathcal{D}_n) \right) + 2\sigma\boldsymbol{\xi}(t)$ with $\langle \xi_i(t)\xi_j(t') \rangle = \delta_{ij}\delta(t-t')$, such that as $t \to \infty$ the DNN parameters $\boldsymbol{\theta}$ will be sampled from the equilibrium Gibbs distribution in parameter space ($p(\boldsymbol{\theta}) \propto \exp(-\frac{y}{2\sigma^2}|\boldsymbol{\theta}|^2 - \mathcal{L}/\sigma^2)$).

Conveniently, letting the weight variance $\sigma^2/\gamma$ scale as the width (as one does in standard He initialization He et al. (2015)), the above Gibbs distribution (taken at $\sigma^2 = 1$ for simplicity) can be rewritten as the following distribution involving only DNN outputs ($f$) Naveh et al. (2021)

$$
p(f|X_n) = e^{-\mathcal{S}[f|X_n]}/\mathcal{Z}(X_n).
\tag{4}
$$

Here $\mathcal{S}$ denotes the so-called "action", given by

$$
\begin{aligned}
\mathcal{S}[f|X_n] = &\frac{1}{2\sigma_\Omega^2} \sum_{\mu=1}^{n_\Omega} (L[f](\boldsymbol{x}_\mu) - \phi(\boldsymbol{x}_\mu))^2 \\
&+ \frac{1}{2\sigma_{\partial\Omega}^2} \sum_{\nu=1}^{n_{\partial\Omega}} (f(\boldsymbol{x}_\nu) - g(\boldsymbol{x}_\nu))^2 + \frac{1}{2} \sum_{\mu=0}^{n} \sum_{\nu=0}^{n} f_\mu [K^{-1}]_{\mu\nu} f_\nu,
\end{aligned}
\tag{5}
$$

with the kernel $K$ denoting the covariance function of the Gaussian prior on $f$, namely

$$K(x,y) = \frac{\int d\boldsymbol{\theta} e^{-\frac{\gamma}{2\sigma^2}|\theta|^2} \hat{f}_\theta(x)\hat{f}_\theta(y)}{\int d\boldsymbol{\theta} e^{-\frac{\gamma}{2\sigma^2}|\theta|^2}} \tag{6}$$

and the distribution's normalization $\mathcal{Z}(X_n) = \int df e^{-\mathcal{S}[f|X_n]}$ is called the "partition function". The first two terms in the action are associated with the loss function, while the last term is associated with the GP prior and makes $f$ vectors which lay in the high eigenvalue of $K$ more likely. Note that, if the variance of the additive noise to the weights during the optimization is $\sigma^2 \neq 1$, then $\sigma_\Omega^2 \rightarrow \sigma_\Omega^2 \sigma^2$ and $\sigma_{\partial\Omega}^2 \rightarrow \sigma_{\partial\Omega}^2 \sigma^2$. Therefore, these constants, $\sigma_\Omega^2$ and $\sigma_{\partial\Omega}^2$, reflect Gaussian uncertainty on the measurement of how well $L[\hat{f}_\theta] = \phi$ and $\hat{f}_\theta = g$, respectively

Finally, we extend the sample index by adding $\mu = 0$ to denote the test point $\boldsymbol{x}_*$ at which we evaluate the estimator, for this reason, the sums in the prior term (third term) in the action start from zero and not one. The test points can be either from the bulk or boundary.

Our task in this work is to derive a systematic and general framework to estimate the value of DNN predictor on a test point $(f(\boldsymbol{x}_*))$ averaged over $p(f|X_n)$ (the posterior) and over draws of $X_n$ from a dataset measure $d\mu_{\mathbf{x}}$.

## 2 OUTLINE OF DERIVATION

The first step, in deriving the NIE, is to remove the randomness induced by the specific choice of bulk and boundary collocation points by averaging over those with a uniform measure. Specifically, we wish to obtain the expectation value of $f(\boldsymbol{x})$ under $p(f|X_n)$ and further average this expectation over all datasets drawn from the same measure. To this end, two tools are used.

**Partition functions and replicas.** Instead of calculating the average $f(\boldsymbol{x})$, we calculate the partition function, defined as $Z[X_n] = \int df e^{-S[f|X_n]}$. We then note that derivatives of $\log(Z[X_n])$ yield various expectation values under the probability $p(f|X_n)$. Consequently, obtaining dataset averages is closely related to calculating the dataset average of $\log(Z[X_n])$. To perform this latter average we use the replica trick, namely we evaluate $Z[X_n]^p$ for an arbitrary positive integer $p$, analytically continue the results to real $p$, using the relation $\log(Z) =_{\lim p \rightarrow 0} (Z^p - 1)/p$, and take the $p \rightarrow 0$ limit.

**Grand-canonical ensemble.** Our next task is to average $Z[X_n]^P$ over all draws of $X_n$. To this end, we find it useful to go from a hard constraint on the number of data points to a soft definition wherein we average overall datasets with a varying number of data points drawn from a Poisson distribution centered around desired $n$. As also studied in Cohen et al. (2021), for large, $n$ the fluctuations introduced by this additional Poisson averaging are insignificant.

Following this route, yields the following continuum action (negative log-probability) for the posterior distribution over network outputs, which becomes exact in the limit of $n_*, \sigma_*^2 \rightarrow \infty$ at fixed $n_*, \sigma_*^2$ (see Appendix B for more details)

$$\mathcal{S}_{\text{eff}}[f] = \frac{n_\Omega}{\sigma_\Omega^2} \int_\Omega dx \, (L[f](\boldsymbol{x}) - \phi(\boldsymbol{x}))^2 + \frac{n_{\partial\Omega}}{\sigma_{\partial\Omega}^2} \int_{\partial\Omega} dx \, (f(\boldsymbol{x}) - g(\boldsymbol{x}))^2$$
$$+ \int_\Omega \int_\Omega f(\boldsymbol{x})[K^{-1}](\boldsymbol{x}, \boldsymbol{y})f(\boldsymbol{y})d\boldsymbol{x}d\boldsymbol{y} \tag{7}$$

where $[K^{-1}]$ is the inverse operator of $K$ namely $\int_\Omega d\boldsymbol{y} K(\boldsymbol{x}, \boldsymbol{y})[K^{-1}](\boldsymbol{y}, \boldsymbol{z}) = \delta(\boldsymbol{x} - \boldsymbol{z})$ for all $\boldsymbol{x}, \boldsymbol{z} \in \Omega$ (including the boundary). We further denote integration with respect to the boundary and bulk measures by $\int_{\partial\Omega}$ and $\int_\Omega$ respectively, such that $\int_\Omega d\boldsymbol{x} = 1$ and, $\int_{\partial\Omega} d\boldsymbol{x} = 1$. Interestingly, unlike in the discrete matrix case Raissi et al. (2017a), the action does not separate boundary-boundary, boundary-bulk, and bulk-bulk kernel terms. Notably the above action is valid both for linear and non-linear operators ($L[f]$). Furthermore, although for qualitative purposes the above asymptotic limit typically described finite systems well Williams & Rasmussen (2006); Cohen et al. (2021), systematic corrections in $1/\sigma_*^2$ could be derived via the methods of Cohen et al. (2021).

Assuming first a linear operator $L$ the resulting posterior is quadratic, and the mean value of $f(\boldsymbol{x})$, signifying the GP's average prediction at a test point $\boldsymbol{x}$, can be obtained by variational methods.

Specifically, one looks for $f_0(\boldsymbol{x})$ so that $\mathcal{S}_{\text{eff}}[f_0(\boldsymbol{x}) + \epsilon\delta f(\boldsymbol{x})]$ is $O(\epsilon^2)$ for any smooth $\delta f(\boldsymbol{x})$ variation. Doing so, however, becomes quite cumbersome and subtle due to the appearance of both bulk and boundary measures together with potentially high-order derivatives coming from $L$. Instead, we find that parametrizing the variation as $\int_\Omega d\boldsymbol{y} K(\boldsymbol{x}, \boldsymbol{y})\delta f(\boldsymbol{y})$ simplifies the deviation. Notably, for the generic case of an invertible, $K(\boldsymbol{x}, \boldsymbol{y})$ this results in the same $f_0(\boldsymbol{x})$.

Turning to a non-linear operator $L[f]$, the above variational approach can be straightforwardly repeated where it would predict the most probable (rather than the mean) output of the network. An example of this is given in Appendix B.3.

## 3  RESULTS

Here we outline our main results. Further details on their derivations are found in the appendix.

### 3.1  THE NEURALLY-INFORMED EQUATION (NIE)

The above variational approach, when applied to the case of a general linear operator resulting analysis leads to an integro-differential equation that we refer to as the neurally-informed equation. To illustrate, consider the general linear $d$-dimensional differential operator $L$ of order $s$. While our formalism similarly applies to nonlinear operators, as discussed in Appendix B.1, in particular, see Appendix B.3 for an example of the application of the formalism to a Nonlinear equation, we proceed to explicitly demonstrate its application for a linear operator to simplify the presentation. For this choice of $L$, given that the kernel prior $K$ is invertible, we derive the following Neurally-Informed Equation (NIE) for the external $f_0 : \Omega \to \mathbb{R}$ which approximates the dataset averaged prediction,

$$f_0(\boldsymbol{x}) + \eta_\Omega \int_\Omega (Lf_0(\boldsymbol{y}) - \phi(\boldsymbol{y})) \left[LK\right](\boldsymbol{y}, \boldsymbol{x})d\boldsymbol{y} + \eta_{\partial\Omega} \int_{\partial\Omega} (f_0(\boldsymbol{y}) - g(\boldsymbol{y}))K(\boldsymbol{y}, \boldsymbol{x})d\boldsymbol{y} = 0. \quad (8)$$

The differential operator $L$ is defined such that it acts on the first component of $K$. Notably, $K$ is now treated as a function of two data-points $\boldsymbol{x}, \boldsymbol{y}$ and not as a matrix. We also defined the quantities $\eta_\Omega = \frac{n_\Omega}{\sigma_\Omega^2}$ and $\eta_{\partial\Omega} = \frac{n_{\partial\Omega}}{\sigma_{\partial\Omega}^2}$ which at large noise can be viewed as the effective amount of data (see also Ref. Cohen et al. (2021); Williams & Rasmussen (2006)). We further denote integration with respect to the boundary and bulk measures by $\int_{\partial\Omega}$ and $\int_\Omega$ respectively. For simplicity, we also consider a uniform collocation point distribution for both the boundary, $1/|\partial\Omega|$, and the bulk, $1/|\Omega|$ Raissi et al. (2019). We further note that the above equation is obtained by applying variational calculus to an energy functional obtained as a leading order expansion in $[L[f](\boldsymbol{x}) - \phi(\boldsymbol{x})]^2/\sigma_\Omega^2$ and $[f(\boldsymbol{x}) - g(\boldsymbol{x})]^2/\sigma_{\partial\Omega}^2$. As such, it is valid when the mean training error is small compared to the observation noise. Perturbative corrections to this limit could be readily obtained using the formalism outlined in the appendix subsection B.1 (see also Ref. Cohen et al. (2021)).

Solving the NIE yields the PINN prediction, averaged over DNN ensembles and draws of $X_n$. In the limit of large $n_\Omega$ and $n_{\partial\Omega}$, the first $f_0(\boldsymbol{x})$ term on the left hand side of Eq. (5) is negligible and, at least for sufficiently smooth and non-singular kernels, one finds that the desired solution of the differential equation also satisfies the original PDE in Eq. (1) For finite $\eta_\Omega, \eta_{\partial\Omega}$, this ceases to be the case and spectral bias starts playing a role. Next, we demonstrate the predictive power of this equation on a 1D toy example.

### 3.2  TOY EXAMPLE

To demonstrate the applicability of our approach, consider the following toy example:

$$\begin{aligned}
\partial_x f(x) &= 0 \quad x \in [0, \infty] \\
f(x) &= g_0 \quad x = 0,
\end{aligned} \quad (9)$$

i.e. a first-order, one-dimensional ordinary differential equation on the half-infinite interval with a constant boundary condition $g_0$ at the origin. Following our use here of an infinite interval, $\eta_\Omega$ should not be understood as the number of points but rather as their density. To match this with

finite ($[0, L]$) interval experiments, we set $n_\Omega = L\eta_\Omega$. Notably, since the solutions decay as one goes far away from zero, this $L$ dependence falls out, as also reflected in the good match obtained with numerics.

Let us consider the following network with two trainable layers acting on $x$

$$f(x) = \sum_{c=1}^{C} a_c \cos(w_c x), \tag{10}$$

with $C$ denoting the number of neurons.

Training this network with the appropriate gradient descent dynamics Naveh et al. (2021); Lee et al. (2018), the relevant kernel for GPR coincides with that of a random network with Gaussian weights $a_c \sim \mathcal{N}(0, 1/(C\sqrt{2\pi l^2}))$ and $w_c \sim \mathcal{N}(0, 1/l^2)$. As shown in Appendix C, this leads to the following kernel

$$K(x, x') = \frac{1}{2\sqrt{2\pi l^2}} \left( e^{-\frac{|x-x'|^2}{2l^2}} + e^{-\frac{|x+x'|^2}{2l^2}} \right). \tag{11}$$

The corresponding neurally-informed equation is given by

$$\int_0^\infty dy \left[ K^{-1}(x, y) - \eta_\Omega \delta(x - y) \partial_y^2 \right] f(y) = -\delta(x) \left[ \eta_{\partial\Omega}(f(0) - g_0) - \eta_\Omega \partial_x f(0) \right], \tag{12}$$

where we have acted on the equation with the inverse of $K(x, y)$, defined by the relation $\int dy K^{-1}(x, y) K(y, z) = \delta(x - z)$. To solve this equation, we first focus on inverting the operator $(K^{-1} + \eta_\Omega \partial_x^2)$ on the left-hand side of the above equation. As shown in App. D this yields the following inverse operator (Green's function)

$$G(x, x') = \frac{1}{\pi} \int_{-\infty}^{+\infty} dk \cos(kx) \cos(kx') \left( e^{+(kl)^2/2} + \eta_\Omega k^2 \right)^{-1}. \tag{13}$$

This integral can either be evaluated numerically or via contour integration. Taking the latter approach reveals an infinite set of simple poles. Yet for large $|x - x'|$ and/or for large $\eta_\Omega$, we numerically observe that a single pole dominates the integral, giving $G(x, x') \approx \frac{\kappa}{2} \left( e^{-\kappa|x-x'|} + e^{-\kappa|x+x'|} \right)$ where $\kappa = \frac{1}{\sqrt{l^2/2+\eta_\Omega}}$ for $\kappa l \ll 1$. While this approximation can be systematically improved by accounting for additional poles, in the numerics presented below we have simply calculated this integral numerically.

Next, we write the neurally-informed equation in the following form

$$f(x') = \int dx G(x', x) \delta(x) \left[ \eta_{\partial\Omega} \left[ g_0 - f(0) \right] + \eta_\Omega \left[ Lf \right](0) \right]. \tag{14}$$

We next define $\Delta = \eta_{\partial\Omega}[g_0 - f(0)] + \eta_\Omega[Lf](0)$, such that $f(x) = G(x, 0)\Delta$. We can now obtain an equation for $\Delta$ via,

$$\Delta = \eta_{\partial\Omega}(g(0) - \Delta G(0, 0)), \tag{15}$$

where we used the fact that $\partial_x G(0, 0) = 0$. Following this we get,

$$\Delta(1 + G(0, 0)\eta_{\partial\Omega}) = \eta_{\partial\Omega} g_0 \tag{16}$$

$$\Delta = \frac{\eta_{\partial\Omega} g_0}{1 + G(0, 0)\eta_{\partial\Omega}},$$

finally, we find

$$f(x \geq 0) = g_0 \frac{\eta_{\partial\Omega}}{1 + G(0, 0)\eta_{\partial\Omega}} G(x, 0). \tag{17}$$

We proceed with testing this numerically by comparing the above results with exact GPR (see Appendix A for more detail), known to be equivalent to Langevin training of the associated network at large $C$ Lee et al. (2018); et al (2021); Naveh et al. (2021). Specifically, we consider $L = 512, l = 1, g_0 = 2.5, \sigma_{\partial\Omega}^2 = 0.01$ and a single boundary point ($n_{\partial\Omega} = 1$) and scan $n_\Omega$

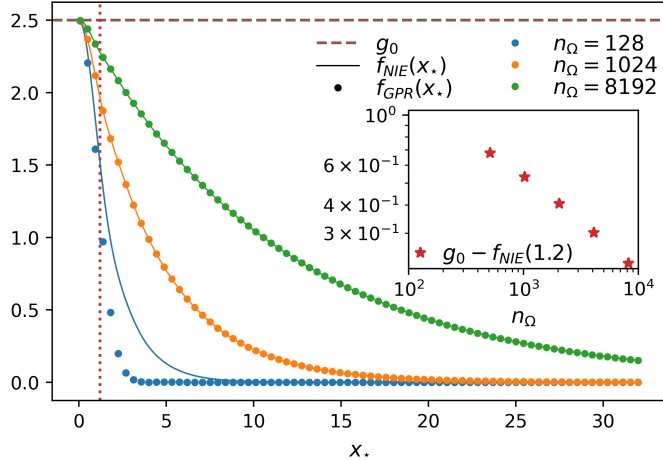

Figure 1: The GPR solution $f_{\text{GPR}}(x_\star)$ (dots) and the neurally-informed equation prediction $f_{\text{NIE}}(x_\star)$ (solid curves) versus $x_\star$ for three values of the number of bulk training points $n_\Omega$. Here $g_0$ (dashed brown curve) denotes the value of the boundary condition at $x_\star = 0$ and, in the toy example, the exact solution for all $x_\star \in \mathbb{R}^+$. We set $L = 2^9 = 512$, $\sigma_\Omega^2 = 2^{-3} = 0.125$, $\ell^2 = 1$, $n_{\partial\Omega} = 1$ and $\sigma_{\partial\Omega}^2 = 2^{-6}/n_\Omega$, such that both $\eta$'s vary with $n_\Omega$ by the same proportion. The vertical dotted red line at $x_\star = 1.2$ is discussed next in the inset. Inset: The difference $g_0 - f_{\text{NIE}}(x_\star = 1.2)$ versus $n_\Omega$ becomes a linear function in log-log scale (red stars). This suggests a power-law scaling, which indicates that the difference vanishes as $n_\Omega \to \infty$.

while keeping the ratio $n_\Omega/\sigma_\Omega^2$ fixed, which means that the neurally-informed equation's predictions remain invariant to $n_\Omega$. In doing so, we are essentially maintaining the amount of "training information" by providing more data while making it noisier. Notably, we do not include any dataset averaging in the GPR numerics. Still, the results agree quite well similar in this aspect to the findings of Cohen et al. (2021). Figure 1 shows the GPR prediction alongside the NIE prediction as a function of the test point's value $x_*$ for $n_\Omega = \{128, 1024, 8192\}$. The figure exhibits an excellent match between theory and experiment, even at relatively small $n_\Omega$.

## 3.3 SPECTRAL BIAS AND FIGURE OF MERIT

A central element in deep learning is finding the optimal DNN architecture for the task at hand. Infinite width limits simplify this process by embodying all of the architecture's details, along with training details such as weight decay terms and gradient noise, in a concrete mathematical object, the kernel $K(\boldsymbol{x}, \boldsymbol{y})$. Qualitatively speaking, these kernels also reflect the useful traits of realistic finite-width DNNs and CNNs Novak et al. (2018).

In the context of standard GPR, the learnability of a particular target function may be analyzed by projection onto the eigenfunctions of the continuum kernel operator Cohen et al. (2021); Canatar et al. (2021); Williams & Rasmussen (2006). The GPR prediction, averaged over the dataset distribution, essentially acts as a high-pass linear filter on these eigenfunctions. This kernel-spectrum-based filtering is what we refer to as spectral bias.

In Appendix (E) we show that a somewhat similar form of spectral bias occurs in PINNs. However rather than being based on the spectrum of $K(\boldsymbol{x}, \boldsymbol{y})$, the relevant eigenvalues $(\hat{\lambda}_k)$ and eigenfunctions $(\varphi_k(\boldsymbol{x}))$ are those obtained by the following diagonalization procedure

$$\hat{K}_{\boldsymbol{x}\boldsymbol{y}} = K(\boldsymbol{x}, \boldsymbol{y}) - \int_{\partial\Omega} \int_{\partial\Omega} K(\boldsymbol{x}, \boldsymbol{z}_1)[K + \eta_{\partial\Omega}^{-1}]^{-1}(\boldsymbol{z}_1, \boldsymbol{z}_2) K(\boldsymbol{z}_2, \boldsymbol{y}) d\boldsymbol{z}_1 d\boldsymbol{z}_2 \tag{18}$$

$$\int_\Omega [L\hat{K}L^\dagger](\boldsymbol{x}, \boldsymbol{y})\varphi_k(\boldsymbol{y}) = \hat{\lambda}_k \varphi_k(\boldsymbol{x}) d\boldsymbol{x},$$

where $\hat{K}$ coincides with the dataset-averaged posterior covariance, given that one introduces only boundary points and fixes them to zero Cohen et al. (2021), $L\hat{K}L^\dagger = L_x L_y K(\boldsymbol{x}, \boldsymbol{y})$ where $L_*$ is

the differential operator acting on the variable $* = \{x, y\}$, and the inverse of $[K + \eta_{\partial\Omega}^{-1}]$ is calculated with respect to the boundary measure. We note in passing that for small $\eta_{\partial\Omega}$, or if the boundary is a set of isolated points, $\hat{K}$ can be expressed straightforwardly (see Appendix E). Next, we find that the discrepancy between $L[f]$ and $\phi$ is given by a low-pass filter on a boundary-augmented source term ($\hat{\phi}$), specifically

$$L[f](\boldsymbol{x}) - \phi(\boldsymbol{x}) = \sum_k \frac{1}{1 + \hat{\lambda}_k \eta_\Omega} c_k \varphi_k(\boldsymbol{x}) \tag{19}$$

$$\hat{\phi}(\boldsymbol{x}) = \phi(\boldsymbol{x}) + \eta_{\partial\Omega} \int_{\partial\Omega} [L\hat{K}](\boldsymbol{x}, \boldsymbol{z}) g(\boldsymbol{z}) d\boldsymbol{z}$$

$$\hat{\phi}(\boldsymbol{x}) = \sum_k c_k \varphi_k(\boldsymbol{x}).$$

The spectral bias presented in Eq. 19 is the second key result of this work. It shows that smaller discrepancies are achieved when the augmented source term ($\hat{\phi}(\boldsymbol{x})$) lays in a high eigenvalues sector of $L\hat{K}L^\dagger$. Furthermore, the spectral components of $\phi(\boldsymbol{x})$ supported on $\hat{\lambda}_k \ll \eta_\Omega^{-1} = \sigma_\Omega^2/n_\Omega$ are effectively filtered out by the PINN. This suggests the following figure of merit measuring the overlap of the discrepancy with the augment source term, namely,

$$Q_n[\phi, g] \equiv \frac{\int_\Omega \hat{\phi}(\boldsymbol{x})[L[f](\boldsymbol{x}) - \phi(\boldsymbol{x})]d\boldsymbol{x}}{\int_\Omega |\hat{\phi}(\boldsymbol{x})|^2 d\boldsymbol{x}} = \frac{\sum_k \frac{|c_k|^2}{1 + \hat{\lambda}_k \eta_\Omega}}{\sum_k |c_k|^2} = \eta_\Omega^{-1} \frac{||\hat{\phi}||^2_{\hat{K} + \eta_\Omega^{-1}}}{\int_\Omega |\hat{\phi}(\boldsymbol{x})|^2 d\boldsymbol{x}}, \tag{20}$$

where $||\hat{\phi}||^2_{\hat{K}+\eta_\Omega^{-1}}$ is the RKHS norm of $\hat{\phi}$ with respect to the kernel $L\hat{K}L^\dagger + \eta_\Omega^{-1}$.

Omitting boundary effects, plausibly relevant to large domains, the above figure of merit has an additional simple interpretation. Indeed, boundary conditions can be viewed as fixing the zero modes of $L$ and are hence directly related to its non-invertibility. Viewing $L$ as an invertible operator is thus consistent with neglecting boundary effects. Doing so, and recalling that for the exact solution $\phi = Lf$, the above figure of merit simplifies to the RKHS norm of the targeted solution ($f$) with respect to $K$ (at large $\eta_\omega$). Notably the same RKHS norm, but with respect to the average predictions, also appears in our continuum action Eq. 36 as a complexity regulator. These two observations suggest that the spectral decomposition of $f$ with respect to $K$ may also serve as an indicator of performance.

## 4 MEASURING SPECTRAL BIAS IN EXPERIMENTS

Here, we demonstrate some potentially practical aspects of our theory by contrasting our spectral bias measure on simple PINN experiments. Specifically, we consider the following cumulative spectral function,

$$A_k[Q, f] = |P_k f|^2/|f|^2 \tag{21}$$

where $P_k$ is the projection of a function $f$ on the $k$ leading eigenfunctions (with respect to the uniform bulk measure) of a kernel $Q$. Notably $A_k$ is a monotonously non-decreasing function, $A_{k=0} = 0$, and generically $A_{k\to\infty} = 1$. Taking $Q = L\hat{K}L^T$ and $f = \hat{\phi}$, a fast-increasing $A_k$ is closely related to the figure of merit in Eq. 19. Taking $Q = K$ and $f$ equal to the exact solution yields an additional perspective related to the second figure of merit, which ignores boundaries. As shown below, we find that one can obtain a numerical approximation for $A_k$ by diagonalizing the kernel as a matrix on a reasonable size grid.

The toy problem we consider is the $1D$ heat equation with a source term,

$$\frac{\partial u}{\partial t} - \frac{\partial^2 u}{\partial x^2} = \frac{e^{-t-\frac{x^2}{2a}}}{a^2} \left[ 2a\pi x \cos(\pi x) + (a + a^2(\pi^2 - 1) - x^2) \sin(\pi x) \right], \tag{22}$$

on the domain $x \in [-1, 1]$ and $t \in [0, 1]$ with initial conditions $u(x, 0) = e^{-\frac{x^2}{2a}} \sin(\pi x)$ and boundary conditions $u(-1, t) = u(1, t) = 0$. The exact solution here is known and given by

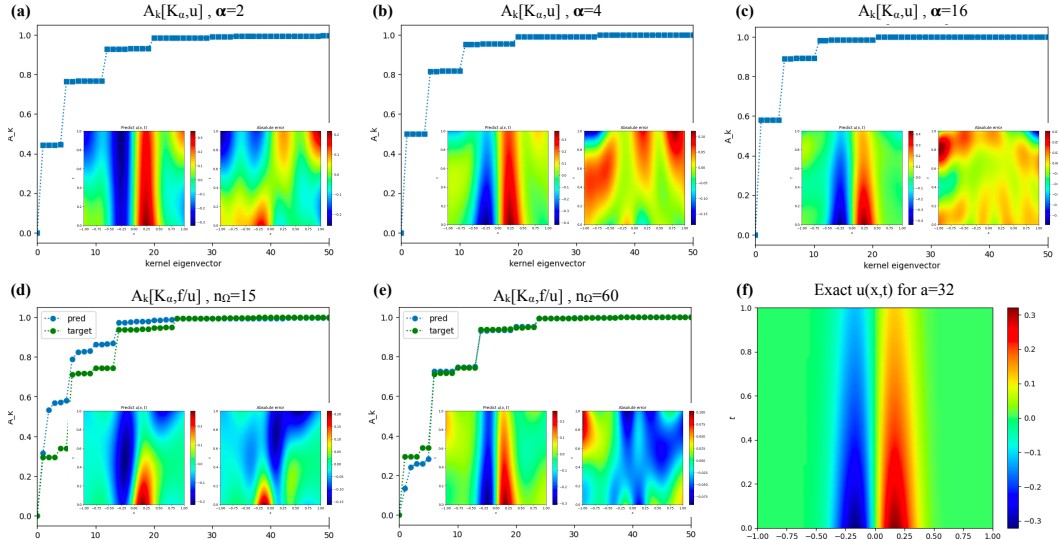

Figure 2: $1D$ heat equation with a source term, solved using the Deepdxe library Lu et al. (2021). Panels (a)-(c): Spectral bias, as measured via the cumulative spectral function, varying only the initialization variances ($\alpha$) between panels. Dominating cumulative spectral functions are indicative of better performance, as shown in the insets. Panels (d-e): Cumulative spectral functions change as a function of the training set size, allowing learning of weaker kernel modes. Panel (f): Exact solution for the PDE used in panels (d-e). The one for (a-c) is the same, but spatially stretched by a factor of $\sqrt{2}$.

$u(x,t) = e^{-t-\frac{x^2}{2a}}\sin(\pi x)$. The constant $a$ evidently determines the spatial scale of the solution and one can expect that very low $a$ values, associated with high-Fourier modes, would be generally more difficult to learn.

In our first experiment, we consider $a = 1/16$ and a student network of the type $\sum_{i=1}^{N} a_i \sin(\boldsymbol{w}_i \cdot \boldsymbol{x})$. We train using the Deepdxe package Lu et al. (2021), use an SGD optimizer at a learning rate of 5e-5, for 150k epochs, and take $N = 512$ and $n_\Omega = n_{\partial\Omega} = 320$. Our control parameter is the normalized standard deviations of the weight initialization, denoted by $\alpha$, which sets the scale of $\sqrt{N}\boldsymbol{w}_i$ and $\sqrt{N}a_i$. We measure $A_k[K_\alpha, u]$, where $K_\alpha$ is the kernel obtained from an infinite-width random DNN with those initialized weights. As shown in Figure 2, taking $\alpha$ between 2 to 16 improves the cumulative spectral functions as well as performance.

Next we consider $a = 1/32$ and a student network of the type $\sum_{i=1}^{N} a_i \text{Erf}(\boldsymbol{w}_i \cdot \boldsymbol{x})$ and fixed $\alpha = 2$. We train again in a similar fashion (SGD with lr $= 1e-5$, epochs $= 160k$) but change the number of data points fixing $n_\Omega = 2n_{\partial\Omega}/3$. We examine $A_k[K_\alpha, \bar{f}]$ where $\bar{f}$ is the average prediction averaged on 32 data and training seeds and contrast it with $A_k[K_\alpha, u]$, where $u$ is the above analytical solution of the equation. As expected, a lower number of data points implies that lower number of kernel eigenvalues participating in predictions.

We comment on other related measures, however, as our focus in this work is mainly theoretical, studying their particular merits is left for future work. One option is plotting $A_k$ as a function of $-\log(\lambda_k)$ instead of $k$ to reflect the difficulty in learning lower spectral modes. Another option is to examine the discrepancy in predictions, namely $A_k[Q, f - u]$. Finally, one can also examine $A_k[LKL^T, \phi]$, where $\phi$ denotes the source term on the right-hand-side of Eq. 22, which has the benefit of depending only on the PDE data and not the solution.

## 5 DISCUSSION

In this work, we laid out an analytical framework for studying overparametrized PINNs. Utilizing a mapping to Gaussian process regression and leveraging tools from statistical mechanics, we provided a concrete analytical formula for their average prediction, in the form of an integro-differential

equation (the Neurally-Informed Equation). Our formalism quantifies spectral bias in PINN prediction and provides concrete figures of merit for kernel-task alignment.

As NIE can be similarly derived for non-linear PDEs (see for instance Appendix B.3), it would be interesting to explore the notion of spectral bias and kernel-task alignment for equations exhibiting strong non-linear effects such as shock-waves Rodriguez-Torrado et al. (2022); Lv et al. (2023); Fuks & Tchelepi (2020). Another direction is adapting our formalism to inverse problems Raissi et al. (2019), i.e. cases in which the coefficients of the equation are learned along with the equation's solution. Finite width effects may also be incorporated into our approach using the methods of Refs. Li & Sompolinsky (2021); Seroussi et al. (2023), as the latter provide effective GPR descriptions of the finite network to which our results readily apply.

A general difficulty in making predictions on DNN performance is the detailed knowledge which is required of the high-dimensional input data distribution. Even for a fixed kernel, this distribution has a strong effect on the kernel spectrum and hence, via spectral bias, on what can be learned efficiently. In contrast, for PINNs, this obstacle is largely avoided as the data distributions are typically low dimensional and often uniform. Furthermore, qualitative properties of the target function (i.e. the desired PDE solution) are often well understood. We thus believe that theory, such as that developed here for spectral bias, is more likely to be a driving force in this sub-field of deep learning.

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

## A GAUSSIAN PROCESS REGRESSION (GPR) FORMULA

Focusing on linear differential operators, and taking the prior on $f$ to be a Gaussian process (GP) with kernel, $K$. As shown in Ref. Raissi et al. (2017b), or in Ref. Pang & Karniadakis (2020) section 14.3.3 for Laplacian operator, the average of $f(\boldsymbol{x}_*)$ under the multivariate Gaussian distribution in Eq. 5 is given by

$$\langle f_* \rangle_{P(f|X_n)} = \vec{k}^T \left( K_{\text{PINN}} + \tilde{\mathrm{I}} \right)^{-1} \boldsymbol{y}, \tag{23}$$

where

$$\boldsymbol{y} = (\boldsymbol{\phi}, \boldsymbol{g})^T, \tag{24}$$

the GP kernel is divided into blocks,

$$K = \left[ \begin{array}{cc} K(X_{n_\Omega}, X_{n_\Omega}) & K(X_{n_{\partial\Omega}}, X_{n_\Omega}) \\ K(X_{n_{\partial\Omega}}, X_{n_\Omega}) & K(X_{n_{\partial\Omega}}, X_{n_{\partial\Omega}}) \end{array} \right] = \left[ \begin{array}{cc} K_{\Omega\Omega} & K_{\partial\Omega\Omega} \\ K_{\Omega\partial\Omega} & K_{\partial\Omega\partial\Omega} \end{array} \right], \tag{25}$$

where $X_{n_\Omega}, X_{n_{\partial\Omega}}$ are the data-points on the domain, $\Omega$ and on the boundary, $\partial\Omega$, respectively. We also define

$$\tilde{\mathrm{I}} = \left( \begin{array}{cc} \sigma_\Omega^2 \mathrm{I} & 0 \\ 0 & \sigma_{\partial\Omega}^2 \mathrm{I} \end{array} \right). \tag{26}$$

where

$$K_{\text{PINN}} = \left( \begin{array}{cc} L K_{\Omega,\Omega} L^\dagger & L K_{\Omega,\partial\Omega} \\ K_{\partial\Omega,\Omega} L^\dagger & K_{\partial\Omega,\partial\Omega} \end{array} \right) \text{ and } \vec{k} = \left( \begin{array}{c} \vec{k}_\Omega L^\dagger \\ \vec{k}_{\partial\Omega} \end{array} \right), \tag{27}$$

where $\left( \vec{k}_\Omega \right)_\mu = K \left( \boldsymbol{x}_*, \boldsymbol{x}_{\mu \in [1, n_\Omega]} \right)$ and $\left( \vec{k}_{\partial\Omega} \right)_\mu = K \left( \boldsymbol{x}_*, \boldsymbol{x}_{\mu \in [n_\Omega + 1, n_\Omega + n_{\partial\Omega}]} \right)$. We denote by $L$ the linear operator acting in the forward direction and $L^\dagger$ the same operator acting "backwards" so that $K L^\dagger$ is $K$ acted upon by $L$ on its second argument.

## B DERIVATION OF THE NEURALLY-INFORMED EQUATION (NIE)

In this section, we derive our main result, the Neurally Informed Equation, Eq. equation 8 in the main text. We start by analyzing posterior distribution in function space and show that it can be written in terms of an effective energy function "action" for a general operator. This provides our first result. We then focus on a linear differential operator and use variational calculus to derive the NIE.

### B.1 DERIVATION OF THE EFFECTIVE ACTION

Using the Bayesian perspective, the posterior distribution in the function space can be written as follows:

$$\begin{aligned} p(f|X_n) &= \mathcal{N}(\mathcal{L}(X_n), \sigma^2 = 1) p_0(f|X) \\ &= \prod_{\mu=1}^{n_\Omega} \mathcal{N}(L[f](\boldsymbol{x}_\mu) - \phi(\boldsymbol{x}_\mu), \sigma_\Omega^2) \prod_{\nu=1}^{n_{\partial\Omega}} \mathcal{N}(f(\boldsymbol{x}_\nu) - g(\boldsymbol{x}_\nu), \sigma_{\partial\Omega}^2) p_0(f|X_n), \end{aligned} \tag{28}$$

where the distribution of the output of the network is given by averaging over the final distribution of the weight $p_0(f|X_n) = \int d\boldsymbol{\theta} p(\boldsymbol{\theta}) \prod_\mu \delta \left( f(\boldsymbol{x}_\mu) - \hat{f}_{\boldsymbol{\theta}}(\boldsymbol{x}_\mu) \right)$, which in the infinite width become Gaussian with kernel $K$ Naveh et al. (2021), $\mathcal{L}(X_n)$ is the training loss in equation 2. Adopting a statistical physics viewpoint, we can write the posterior distribution as $p(f|X_n) = \frac{1}{\mathcal{Z}[X_n]} e^{-\mathcal{S}[f|X_n]}$, where

$$\mathcal{S}[f|X_n] = \mathcal{S}_{\text{PDE}}[f|X_n] + \mathcal{S}_0[f|X_n] \tag{29}$$

where,

$$\mathcal{S}_{\mathrm{PDE}}[f|X_n] = \mathcal{S}_\Omega[f|X_n] + \mathcal{S}_{\partial\Omega}[f|X_n] \tag{30}$$

$$= \frac{1}{2\sigma_\Omega^2} \sum_{\mu=1}^{n_\Omega} \left(L[f](\boldsymbol{x}_\mu) - \phi(\boldsymbol{x}_\mu)\right)^2 + \frac{1}{2\sigma_{\partial\Omega}^2} \sum_{\nu=1}^{n_{\partial\Omega}} \left(f(\boldsymbol{x}_\nu) - g(\boldsymbol{x}_\nu)\right)^2 \tag{31}$$

$$\mathcal{S}_0[f|X_n] = \frac{1}{2} \sum_{\mu=1}^{n} \sum_{\nu=1}^{n} f_\mu [K^{-1}]_{\mu\nu} f_\nu + \frac{1}{2} \log|K| + \frac{n}{2} \left(3\log 2\pi + 2\log \sigma_{\partial\Omega}\sigma_\Omega\right), \tag{32}$$

and $\mathcal{Z}[X_n] = \int e^{-\mathcal{S}[f|X_n]} df$ is the partition function.

To understand generalization, we are interested in computing ensemble averages over many data-sets. In the following, we use our freedom to choose the prior on the function to be on an infinite dataset, i.e. in the reproduction kernel Hilbert space (RKHS), with the kernel being the continuum version of Eq. (25) Williams & Rasmussen (2006). Utilizing a statistical mechanics approach, we want to compute the free energy, $\mathbb{E}_{\mathcal{D}_n}[\log\mathcal{Z}[X_n]]$ which can also be viewed as the generating function of the process. For this purpose, we employ the replica trick:

$$\mathbb{E}_{\mathcal{D}_n}\left[\log\mathcal{Z}[X_n]\right] = \lim_{p\to 0} \frac{\partial \log \mathbb{E}_{\mathcal{D}_n}\left[\mathcal{Z}^p[X_n]\right]}{\partial p} \tag{33}$$

where

$$\mathbb{E}_{\mathcal{D}_n}\left[\mathcal{Z}^p[X_n]\right] = \mathbb{E}_{\mathcal{D}_n}\left[\int \prod_\alpha^p e^{-\mathcal{S}[f_\alpha|X_n]} df_\alpha\right]$$

$$= \int \mathbb{E}_{\mathcal{D}_n}\left[e^{-\sum_{\alpha=1}^p \mathcal{S}_{\mathrm{PDE}}[f_\alpha|X_n]}\right] \prod_\alpha^p p_0(f_\alpha) df_\alpha$$

$$= \langle \mathbb{E}_{\mathcal{D}_n}\left[e^{-\sum_{\alpha=1}^p \mathcal{S}_{\mathrm{PDE}}[f_\alpha|X_n]}\right]\rangle_{0,p} = \left\langle \left(\int_\Omega e^{-\sum_{\alpha=1}^p \mathcal{S}_\Omega[f_\alpha|\boldsymbol{x}]} d\boldsymbol{x}\right)^{n_\Omega} \left(\int_{\partial\Omega} e^{-\sum_{\alpha=1}^p \mathcal{S}_{\partial\Omega}[f_\alpha|\boldsymbol{x}]} d\boldsymbol{x}\right)^{n_{\partial\Omega}} \right\rangle_{0,p} \tag{34}$$

where $\langle...\rangle_{0,p}$ denotes expectation over the $p$ replicated prior distribution, $p_0(f)$ over the continuum measure over $\Omega$ with kernel $K$, where $\alpha$ denotes the index of the replica modes of the system, in addition to the above we use the fact that the samples are i.i.d. The above object is still hard to analyze due to the coupling between the replica modes. To make progress, we follow a similar approach as in Cohen et al. (2021); Malzahn & Opper (2001). We transform into a "grand canonical" partition function, meaning that we treat the dataset size $n$ as a random variable drawn from a Poisson distribution which we average over. The idea is that for a large enough data set size, $n = n_{\partial\Omega} + n_\Omega$, the dominating term in the grand canonical partition function $\bar{G}_p$ is a good estimator of the entire free energy. Alternatively stated, we compute here the average over all different datasets of size $n$ where $n$ itself is drawn from a Poisson distribution whose average is $\bar{n}$. We expect this average quantity to coincide with averaging using a fixed $n = \bar{n}$ at large $n$ (see also Ref. Cohen et al. (2021) for a demonstration of this).

The grand canonical partition function is given by

$$\bar{G}_p = \mathbb{E}_n \left[ \mathbb{E}_{\mathcal{D}_n} \left[ \mathcal{Z}^p[X_n] \right] \right]$$

$$= \langle \sum_{u_1=0}^{\infty} \frac{\bar{\eta}_{\Omega}^{u_1} e^{-\bar{\eta}_{\Omega}}}{u_1!} \mathbb{E}_{\mathcal{D}_{u_1}} \left[ e^{-\sum_{\alpha=1}^{p} \mathcal{S}_{\Omega}[f_\alpha | X_{u_1}]} \right] \sum_{u_2=0}^{\infty} \frac{\bar{\eta}_{\partial\Omega}^{u_2} e^{-\bar{\eta}_{\partial\Omega}}}{u_2!} \mathbb{E}_{\mathcal{D}_{u_2}} \left[ e^{-\sum_{\alpha=1}^{p} \mathcal{S}_{\partial\Omega}[f_\alpha | X_{u_2}]} \right] \rangle_{0,p}$$

$$= \langle \sum_{u_1=0}^{\infty} \frac{\bar{\eta}_{\Omega}^{u_1} e^{-\bar{\eta}_{\Omega}}}{u_1!} \left( \int_{\Omega} e^{-\sum_{\alpha=1}^{p} \mathcal{S}_{\Omega}[f_\alpha | \boldsymbol{x}]} d\boldsymbol{x} \right)^{u_1} \sum_{u_2=0}^{\infty} \frac{\bar{\eta}_{\partial\Omega}^{u_2} e^{-\bar{\eta}_{\partial\Omega}}}{u_2!} \left( \int_{\partial\Omega} e^{-\sum_{\alpha=1}^{p} \mathcal{S}_{\partial\Omega}[f_\alpha | \boldsymbol{x}]} d\boldsymbol{x} \right)^{u_2} \rangle_{0,p}$$

$$= \int \prod_{\beta}^{p} e^{-\bar{\eta} + \bar{\eta}_{\Omega} \mathbb{E}_{\boldsymbol{x}} \left[ e^{-\sum_{\alpha=1}^{p} \mathcal{S}_{\Omega}[f_\alpha | \boldsymbol{x}]} \right] + \bar{\eta}_{\partial\Omega} \mathbb{E}_{\boldsymbol{x}} \left[ e^{-\sum_{\alpha=1}^{p} \mathcal{S}_{\partial\Omega}[f_\alpha | \boldsymbol{x}]} \right]} p_0(f_\beta) df_\beta$$

$$= \int e^{-\mathcal{S}_{\text{eff}}[\{f_\alpha\}_{\alpha=1}^{p}]} \prod_{\beta}^{p} df_\beta \quad (35)$$

where in the third transition, we use the fact all points are chosen to be independent and identically distributed. With a slight abuse of notation, we take the number of data points in the bulk $n_{\Omega} \sim \text{Pois}(\bar{\eta}_{\Omega})$, and on the boundary $n_{\partial\Omega} \sim \text{Pois}(\bar{\eta}_{\partial\Omega})$ as independent Poisson-distributed random variables. We denote by $\bar{\eta} = \bar{\eta}_{\Omega} + \bar{\eta}_{\partial\Omega}$, where $\bar{\eta}_{\Omega}$ and $\bar{\eta}_{\partial\Omega}$, the expectation of the Poisson variables on the bulk and boundary, are the true deterministic number of points in the bulk and on the boundary, respectively. The effective action is then defined as:

$$\mathcal{S}_{\text{eff}}[\{f_\alpha\}_{\alpha=1}^{p}] = \bar{\eta} - \bar{\eta}_{\Omega} \int_{\Omega} e^{-\sum_{\alpha=1}^{p} \mathcal{S}_{\Omega}[f_\alpha | \boldsymbol{x}]} d\boldsymbol{x} - \bar{\eta}_{\partial\Omega} \int_{\partial\Omega} e^{-\sum_{\alpha=1}^{p} \mathcal{S}_{\partial\Omega}[f_\alpha | \boldsymbol{x}]} d\boldsymbol{x}$$

$$+ \frac{1}{2} \sum_{\alpha=1}^{p} \int_{\Omega} \int_{\Omega} f_\alpha(\boldsymbol{x})[K^{-1}](\boldsymbol{x}, \boldsymbol{y}) f_\alpha(\boldsymbol{y}) d\boldsymbol{y} d\boldsymbol{x}. \quad (36)$$

Next, Taylor expanding the above exponent, we obtain the following effective action in first-order for each replica mode $\alpha \in [1, p]$:

$$\mathcal{S}_{\text{eff}}[f_\alpha] = \bar{\eta}_{\Omega} \int_{\Omega} \mathcal{S}_{\Omega}[f_\alpha | \boldsymbol{x}] d\boldsymbol{x} - \bar{\eta}_{\partial\Omega} \int_{\partial\Omega} \mathcal{S}_{\partial\Omega}[f_\alpha | \boldsymbol{x}] d\boldsymbol{x}$$

$$+ \frac{1}{2} \int_{\Omega} \int_{\Omega} f_\alpha(\boldsymbol{x})[K^{-1}](\boldsymbol{x}, \boldsymbol{y}) f_\alpha(\boldsymbol{y}) d\boldsymbol{x} d\boldsymbol{y} + O(\bar{\eta} S_{\text{PDE}}^2), \quad (37)$$

where $\sigma^2 = \sigma_{\Omega}^2 + \sigma_{\partial\Omega}^2$. Note that, the first-order action decouples the replica modes, which drastically simplifies the analysis. Since all replicas are now decoupled one can take the replica limit (see equation 33) and obtain

$$\lim_{p \to 0} \frac{\partial \log \bar{G}_p}{\partial p} = \lim_{p \to 0} \frac{\partial \log \mathbb{E}_n[\mathbb{E}_{\mathcal{D}_n}[\mathcal{Z}^p]]}{\partial p} = \lim_{p \to 0} \frac{\partial \log[\mathbb{E}_n[\mathbb{E}_{\mathcal{D}_n}[\mathcal{Z}]]]^p}{\partial p} = \lim_{p \to 0} \log \bar{G} = \log \bar{G},$$

where $\bar{G} = \int e^{-\mathcal{S}_{\text{eff}}[f]} df$. We comment that taking second-order corrections in $\mathcal{S}_{\Omega/\partial\Omega}[f_\alpha | \boldsymbol{x}]$ into account leads to a tighter prediction Cohen et al. (2021).

Figure 1 shows that the estimated output derived using the first-order action provides a good prediction for a large amount of data. For an analysis of the higher-order terms in the context of regression, see Cohen et al. (2021). Using variational calculus, this action provides the network's average prediction at any point $\boldsymbol{x}$ in the domain for a general nonlinear operator. Intuitively, one would expect that the effective action follows from taking the continuum limit of $\mathcal{S}[f | X_n]$. Yet doing so is subtle, due to the appearance of separate bulk and boundary measures in the continuum limit and the fact that the operator $K^{-1}$ (unlike $K$) is measure-dependent. More precisely, the inversion operation depends on the whole measure of points both on the boundary and on the bulk $\mu(\boldsymbol{x})$ meaning that $\int [K]^{-1}(\boldsymbol{z}, \boldsymbol{x}) K(\boldsymbol{x}, \boldsymbol{y}) \mu(\boldsymbol{x}) d\boldsymbol{x} = \delta_{\boldsymbol{z}, \boldsymbol{y}} / \mu(y)$.

In the next section, we apply the calculus of variations to derive the neurally-informed equation (NIE) for the estimator from the effective action.

## B.2 DERIVATION OF THE NIE USING VARIATIONAL CALCULUS

One could apply variational calculus to derive from the effective action an NIE for a general non-linear operator. However, since there are infinitely many ways for the operator to be nonlinear (e.g. $L = L(1, f, f_x, f_{xx}, .., f^2, f_x f, ...)$), and this is very much problem dependent. Here we derive the NIE for a general linear operator, $L$ and discuss a particular non-linear operator below. The operator $L$ is applied to the scalar function $f(\boldsymbol{x})$, with $\boldsymbol{x} \in \mathbb{R}^d$. The corresponding effective action is

$$\mathcal{S}_{\text{eff}}[f] = \frac{1}{2}\eta_\Omega \int_\Omega \left(Lf(\boldsymbol{x}) - \phi(\boldsymbol{x})\right)^2 d\boldsymbol{x} + \frac{1}{2}\eta_{\partial\Omega} \int_{\partial\Omega} \left(f(\boldsymbol{x}) - g(\boldsymbol{x})\right)^2 d\boldsymbol{x}$$
$$+ \frac{1}{2}\int_\Omega \int_\Omega f(\boldsymbol{x})[K^{-1}](\boldsymbol{x}, \boldsymbol{y})f(\boldsymbol{y})d\boldsymbol{y}d\boldsymbol{x}. \quad (38)$$

Suppose that $f_0$ is a minimizer, and $h : \Omega \to \mathbb{R}$ is a variation function,

$$\delta\mathcal{S}_{\text{eff}}[f_0] = \frac{1}{2}\frac{d}{d\epsilon}\mathcal{S}_{\text{eff}}(f_0 + \epsilon h, Lf_0 + \epsilon Lh)) \mid_{\epsilon=0}$$

$$= \eta_\Omega \int_\Omega \left(Lf_0(\boldsymbol{x}) + \epsilon Lh(\boldsymbol{x}) - \phi(\boldsymbol{x})\right) Lh(\boldsymbol{x})d\boldsymbol{x}$$

$$+ \eta_{\partial\Omega} \int_{\partial\Omega} \left(f_0(\boldsymbol{x}) - \epsilon h(\boldsymbol{x}) - g(\boldsymbol{x})\right) h(\boldsymbol{x})d\boldsymbol{x}$$

$$+ \frac{1}{2}\int_\Omega h(\boldsymbol{x})\int_\Omega [K^{-1}](\boldsymbol{x}, \boldsymbol{y})\left(f_0(\boldsymbol{y}) + \epsilon h(\boldsymbol{y})\right) d\boldsymbol{x}d\boldsymbol{y}$$

$$+ \frac{1}{2}\int_\Omega \left(f_0(\boldsymbol{x}) + \epsilon h(\boldsymbol{x})\right)\int_\Omega [K^{-1}](\boldsymbol{x}, \boldsymbol{y})]h(\boldsymbol{y})d\boldsymbol{y}d\boldsymbol{x} \mid_{\epsilon=0}$$

$$= \eta_\Omega \int_\Omega d\boldsymbol{x} \left(Lf_0(\boldsymbol{x}) - \phi(\boldsymbol{x})\right) Lh(\boldsymbol{x})$$

$$+ \eta_{\partial\Omega} \int_{\partial\Omega} \left(f_0(\boldsymbol{x}) - g(\boldsymbol{x})\right) h(\boldsymbol{x})d\boldsymbol{x}$$

$$+ \int_\Omega \int_\Omega [K^{-1}](\boldsymbol{x}, \boldsymbol{y})f_0(\boldsymbol{x})h(\boldsymbol{y})d\boldsymbol{y}d\boldsymbol{x} \quad (39)$$

Next, we use our freedom to parameterize the variation to simplify the boundary terms. Specifically, we consider a variation of the form

$$h(\boldsymbol{x}) = (K \star \tilde{h})(\boldsymbol{x}) = \int_\Omega K(\boldsymbol{x}, \boldsymbol{y})\tilde{h}(\boldsymbol{y})d\boldsymbol{y}, \quad (40)$$

where $\star$ denote the convolution operator on the bulk measure and $\tilde{h} : \Omega \to \mathbb{R}$. Provided that the kernel $K$ is invertible, we stress that any variation can be presented in this manner. Substituting into Eq. equation 44

$$\delta\mathcal{S}_{\text{eff}}[f_0] = \eta_\Omega \int_\Omega \int_\Omega \left(Lf_0(\boldsymbol{x}) - \phi(\boldsymbol{x})\right) [LK](\boldsymbol{x}, \boldsymbol{y})\tilde{h}(\boldsymbol{y})d\boldsymbol{y}d\boldsymbol{x}$$

$$+ \eta_{\partial\Omega} \int_\Omega d\boldsymbol{y} \int_{\partial\Omega} d\boldsymbol{x}(f_0(\boldsymbol{x}) - g(\boldsymbol{x}))K(\boldsymbol{x}, \boldsymbol{y})\tilde{h}(\boldsymbol{y}) + \int_\Omega f_0(\boldsymbol{x})\tilde{h}(\boldsymbol{x})d\boldsymbol{x}. \quad (41)$$

In the notation, $LK$, the operator $L$ is defined to act on the first coordinate of the operator $K$. Differentiating with respect to $\tilde{h}$ gives

$$\eta_\Omega \int_\Omega \left(Lf_0(\boldsymbol{y}) - \phi(\boldsymbol{y})\right) [LK](\boldsymbol{y}, \boldsymbol{x})d\boldsymbol{y}$$

$$+ \eta_{\partial\Omega} \int_{\partial\Omega} \left(f_0(\boldsymbol{y}) - g(\boldsymbol{y})\right)K(\boldsymbol{y}, \boldsymbol{x})d\boldsymbol{y} + f_0(\boldsymbol{x}) = 0. \quad (42)$$

We note in passing that the above derivation can also be done for a nonlinear operator, by allowing the functional derivative to also act on the operator $L$.

### B.3 NIE FOR NONLINEAR EQUATION - EXAMPLE

Turning to non-linear $L$, we consider, as a pedagogical example, the Fisher equation for Gene propagation, given by the following nonlinear equation

$$\partial_t f(\boldsymbol{x}) = f(\boldsymbol{x})(1 - f(\boldsymbol{x})) + \partial_{xx} f(\boldsymbol{x})$$

where $\boldsymbol{x} = [t, x]$. In this case, one can write $L[f] = (\partial_t - 1 + f - \partial_{xx})f$ with $\phi = 0$, let us take also $g(\boldsymbol{x}) = 1$. The action in this case is then:

$$\mathcal{S}_{\text{eff}}[f] = \frac{1}{2}\eta_\Omega \int_\Omega (L[f](\boldsymbol{x}))^2 \, d\boldsymbol{x} + \frac{1}{2}\eta_{\partial\Omega} \int_{\partial\Omega} (f(\boldsymbol{x}) - g(\boldsymbol{x}))^2 \, d\boldsymbol{x}$$

$$+ \frac{1}{2}\int_\Omega \int_\Omega f(\boldsymbol{x})[K^{-1}](\boldsymbol{x},\boldsymbol{y})f(\boldsymbol{y})d\boldsymbol{y}d\boldsymbol{x}. \quad (43)$$

This yields the variation:

$$\delta\mathcal{S}_{\text{eff}}[f] = \frac{1}{2}\frac{d}{d\epsilon}S_{\text{eff}}(f + \epsilon h, \partial_t f + \epsilon\partial_t h, \partial_{xx} f + \epsilon\partial_{xx} h)) \mid_{\epsilon=0}$$

$$= \eta_\Omega \int_\Omega (\partial_t f(\boldsymbol{x}) - f(\boldsymbol{x}) + f^2(\boldsymbol{x}) - \partial_{xx} f(\boldsymbol{x}))(\partial_t h(\boldsymbol{x}) - h(\boldsymbol{x}) + 2f(\boldsymbol{x})h(\boldsymbol{x}) - \partial_{xx} h(\boldsymbol{x}))d\boldsymbol{x}$$

$$+ \eta_{\partial\Omega} \int_{\partial\Omega} (f(\boldsymbol{x}) - g(\boldsymbol{x}))\, h(\boldsymbol{x})d\boldsymbol{x}$$

$$+ \int_\Omega \int_\Omega [K^{-1}](\boldsymbol{x},\boldsymbol{y})f(\boldsymbol{x})h(\boldsymbol{y})d\boldsymbol{y}d\boldsymbol{x} \quad (44)$$

After re-expressing the variation in terms of the $\tilde{h}(x)$ variation (as in equation 40), we obtain that:

$$\delta\mathcal{S}_{\text{eff}}[f] = \eta_\Omega \int_\Omega \int_\Omega L[f](\partial_t K(\boldsymbol{x},\boldsymbol{y}) - K(\boldsymbol{x},\boldsymbol{y}) + 2f(\boldsymbol{x})K(\boldsymbol{x},\boldsymbol{y}) - \partial_{xx} K(\boldsymbol{x},\boldsymbol{y}))\tilde{h}(\boldsymbol{y})d\boldsymbol{x}d\boldsymbol{y}$$

$$+ \eta_{\partial\Omega} \int_\Omega d\boldsymbol{y} \int_{\partial\Omega} d\boldsymbol{x}(f(\boldsymbol{x}) - g(\boldsymbol{x}))K(\boldsymbol{x},\boldsymbol{y})\tilde{h}(\boldsymbol{y}) + \int_\Omega f(\boldsymbol{x})\tilde{h}(\boldsymbol{x})d\boldsymbol{x}, \quad (45)$$

where, $\partial_{xx} K(\boldsymbol{x},\boldsymbol{y})$ is a second partial derivative with respect to the second argument,$x$, of the first variable $\boldsymbol{x} = [t, x]$. Yielding the following NIE

$$\eta_\Omega \int_\Omega L[f](\partial_{\boldsymbol{z}_0} K(\boldsymbol{z},\boldsymbol{x}) - K(\boldsymbol{z},\boldsymbol{x}) + 2f(\boldsymbol{z})K(\boldsymbol{z},\boldsymbol{y}) - \partial_{\boldsymbol{z}_1\boldsymbol{z}_1} K(\boldsymbol{z},\boldsymbol{x}))d\boldsymbol{z}$$

$$+ \eta_{\partial\Omega} \int_\Omega d\boldsymbol{y} \int_{\partial\Omega} (f(\boldsymbol{y}) - g(\boldsymbol{y}))K(\boldsymbol{y},\boldsymbol{x}) + f(\boldsymbol{x}) = 0. \quad (46)$$

where $\boldsymbol{z}_0$ ($\boldsymbol{z}_1$) refers to the first (second) component of $\boldsymbol{z}$ being time (space)

## C GENERATING SOME NNGP KERNELS

The kernel used in the toy model could be generated using the following random neural network acting on a one-dimensional input $x$

$$f(x) = \sum_{c=1}^C a_c \cos(w_c x) \quad (47)$$

with

$$a_c \sim N(0, \sigma_a^2/C) \quad (48)$$
$$w_c \sim N(0, \sigma_w^2)$$

where we will soon take $\sigma_w^2 = 1/l^2$ and $\sigma_a^2 = \frac{1}{\sqrt{2\pi l^2}}$. The NNGP kernel is then

$$K(x,y) \equiv \langle f(x)f(y)\rangle_{a,w} = \sigma_a^2 \frac{1}{\sqrt{2\pi\sigma_w^2}} \int dw e^{-\frac{w^2}{2\sigma_w^2}} \cos(wx)\cos(wy) \quad (49)$$

$$= \frac{\sigma_a^2}{2\sqrt{2\pi\sigma_w^2}} \int dw e^{-\frac{w^2}{2\sigma_w^2}} [e^{iw(x+y)} + e^{iw(x-y)}] = \frac{\sigma_a^2}{2}[e^{-\frac{\sigma_w^2(x+y)^2}{2}} + e^{-\frac{\sigma_w^2(x-y)^2}{2}}]$$

hence, as mentioned, choosing $\sigma_w^2 = 1/l^2$ and $\sigma_a^2 = 1/\sqrt{2\pi l^2}$ reproduces the desired NNGP kernel. This means that training such a neural network with weight decay proportional to $l^2$ on the $w_c$ weights, PINN loss, and using Langevin type training, samples from the GP posterior we used in our toy example.

## D GREEN'S FUNCTION FOR THE TOY MODEL

We first focus on inverting the bulk operator on the l.h.s. $(K^{-1} - \eta_\Omega \delta(x-y)\partial_y^2)$, which now derive. Consider the following set of basis functions for the positive half-interval,

$$\langle x|k > 0 \rangle = \frac{\sqrt{2}}{\sqrt{\pi}} \cos(kx) = \frac{1}{\sqrt{2\pi}}[e^{ikx} + e^{-ikx}] \tag{50}$$

Notably

$$\langle k|k' \rangle = \int_0^\infty dx \langle k|x \rangle \langle x|k' \rangle \tag{51}$$

$$= \int_0^\infty dx \frac{2}{\pi} \cos(kx) \cos(k'x) = \int_{-\infty}^\infty dx \frac{1}{\pi} \cos(kx) \cos(k'x)$$

$$= \int_{-\infty}^\infty dx \frac{1}{4\pi}[e^{ikx} + e^{-ikx}][e^{ik'x} + e^{-ik'x}] = \delta(|k| - |k'|)$$

Consider $K(x, y)$ on this basis

$$K|k\rangle = \int_0^\infty dy K(x,y) \frac{\sqrt{2}}{\sqrt{\pi}} \cos(ky) = \int_{-\infty}^\infty dy K(x,y) \frac{1}{\sqrt{2\pi}} \cos(ky) \tag{52}$$

$$= \int_{-\infty}^\infty dy K(x,y) \frac{1}{2\sqrt{2\pi}}[e^{iky} + e^{-iky}] = 2e^{-(kl)^2/2} \frac{1}{\sqrt{2\pi}} \cos(kx) = e^{-(kl)^2/2}|k\rangle$$

Transforming the bulk operator to this "$k$-space"

$$\langle k|[K^{-1} - \eta_\Omega \partial_x^2]|k' \rangle = \delta(k - k') \left[ e^{(kl)^2/2} + \eta_\Omega k^2 \right]. \tag{53}$$

We thus find the following expression for the Green function which is defined here as $[K^{-1} + \eta_\Omega L^T L]G(x,y) = \delta(x-y)$,

$$G(x, x') = \frac{2}{\pi} \int_0^\infty dk dk' \cos(kx) \cos(k'x') \delta(k - k') \left[ e^{+(kl)^2/2} + \eta_\Omega k^2 \right]^{-1} \tag{54}$$

$$= \frac{2}{\pi} \int_0^{+\infty} dk \cos(kx) \cos(kx') \left[ e^{+(kl)^2/2} + \eta_\Omega k^2 \right]^{-1}$$

$$= \frac{1}{\pi} \int_{-\infty}^{+\infty} dk \cos(kx) \cos(kx') \left[ e^{+(kl)^2/2} + \eta_\Omega k^2 \right]^{-1}$$

$$= \frac{1}{4\pi} \int_{-\infty}^{+\infty} dk [e^{ikx} + e^{-ikx}][e^{ikx'} + e^{-ikx'}] \left[ e^{+(kl)^2/2} + \eta_\Omega k^2 \right]^{-1}.$$

This integral can be evaluated numerically or via contour integration. Considering the latter, one finds an infinite set of simple poles. As $|x - x'|$ grows or for very large $\eta_\Omega$, we find numerically that a single pole dominates the result and yields $G(x, x') \approx \frac{\kappa}{2} \left[ e^{-\kappa|x-x'|} + e^{-\kappa|x+x'|} \right]$ where $\kappa = \frac{1}{\sqrt{l^2/2 + \eta_\Omega}}$ for $\kappa l \ll 1$. While we believe this approximation can be systematically improved by accounting for additional poles, in the numerics carried below we simply calculate this integral numerically.

## E DERIVING THE $Q_n[\phi, g]$ FIGURE OF MERIT

Here, we derive the $Q_n[\phi, g]$ figure of merit, which also exposes some hidden algebraic relations between the GPR formula and the neurally-informed equation.

First, we note that one can re-rewrite the neurally-informed equation as an operator equation without performing any integration by parts, namely

$$\int_\Omega [K^{-1} + \eta_{\partial\Omega}\delta\delta + \eta_\Omega L^\dagger L]_{\boldsymbol{xy}} f(\boldsymbol{y})d\boldsymbol{y} = \eta_{\partial\Omega}\delta_{\partial\Omega}(\boldsymbol{x})g(\boldsymbol{x}) + \eta_\Omega L^\dagger \phi(\boldsymbol{x}) \tag{55}$$

where $\delta_{\partial\Omega}(\boldsymbol{x}) = \int_{\partial\Omega}\delta(\boldsymbol{x}-\boldsymbol{z})d\boldsymbol{z}$, $[\delta\delta]_{\boldsymbol{xy}} = \int_{\partial\Omega}\delta(\boldsymbol{x}-\boldsymbol{z})\delta(\boldsymbol{z}-\boldsymbol{y})d\boldsymbol{z}$, and $[fL^\dagger]_{\boldsymbol{x}} = [Lf]_{\boldsymbol{x}}$, which is then consistent with $\int_\Omega [L^\dagger L]_{\boldsymbol{xy}} f(\boldsymbol{y})d\boldsymbol{y} = \int_\Omega L_{\boldsymbol{x}}^\dagger \delta(\boldsymbol{x}-\boldsymbol{y})L_{\boldsymbol{y}} f(\boldsymbol{y})d\boldsymbol{y}$. Noting that the operator on the l.h.s. is a sum of positive semi-definite operators and that $K^{-1}$ is positive definite (indeed $K$ is generically semi-definite and bounded), we invert the operator on the left-hand side and obtain

$$f(\boldsymbol{x}) = \eta_{\partial\Omega}\int_{\partial\Omega} \left[[K^{-1} + \eta_\Omega L^\dagger L + \eta_{\partial\Omega}\delta\delta]^{-1}\right]_{\boldsymbol{xz}} g(\boldsymbol{z})d\boldsymbol{z} \tag{56}$$
$$+ \eta_\Omega \int_\Omega \left[[K^{-1} + \eta_\Omega L^\dagger L + \eta_{\partial\Omega}\delta\delta]^{-1}\right]_{\boldsymbol{xy}} [L^\dagger\phi]_{\boldsymbol{y}}d\boldsymbol{y}$$

where the inverse is taken with respect to the bulk measure. Next, we define the following operator,

$$\hat{K}_{\boldsymbol{xy}} \equiv \left[[K^{-1} + \eta_{\partial\Omega}\delta\delta]^{-1}\right]_{\boldsymbol{xy}} \tag{57}$$
$$= K(\boldsymbol{x},\boldsymbol{y}) - \int_{\partial\Omega}\int_{\partial\Omega} K(\boldsymbol{x},\boldsymbol{z}_1)[K + \eta_{\partial\Omega}^{-1}]^{-1}(\boldsymbol{z}_1,\boldsymbol{z}_2)K(\boldsymbol{z}_2,\boldsymbol{y})d\boldsymbol{z}_1 d\boldsymbol{z}_2$$

where the second transition is since the operator $K$ is assumed invertible, a Woodbury-type manipulation can be applied. Note also that the inverse after the second inequality is w.r.t. the boundary measure and $\eta_{\partial\Omega} = n_{\partial\Omega}/\sigma_{\partial\Omega}^2$. If the boundary is a single point, obtaining $\hat{K}$ is again straightforward, since the operator inverse becomes just a simple algebraic inverse. Substituting Eq. equation 57

$$[K^{-1} + \eta_{\partial\Omega}\delta\delta + \eta_\Omega L^\dagger L]^{-1} = [\hat{K}^{-1} + \eta_\Omega L^\dagger L]^{-1}. \tag{58}$$

Next, we perform a similar manipulation to the one leading to $\hat{K}$ namely

$$[\hat{K}^{-1} + \eta_\Omega L^\dagger L]^{-1} = \hat{K}[1 + \eta_\Omega L^\dagger L\hat{K}]^{-1} \tag{59}$$
$$= \hat{K} - \hat{K}L^\dagger \left(\eta_\Omega^{-1} + L\hat{K}L^\dagger\right)^{-1} L\hat{K}$$

To obtain the result in the main text, we apply the operator $L$ on Eq. equation 56, i.e. $Lf$. It has two contributions, we first start with the source term contribution

$$\eta_\Omega L \int_\Omega \left[[\hat{K}^{-1} + \eta_\Omega L^\dagger L]^{-1}\right]_{\boldsymbol{xy}} [L^\dagger\phi]_{\boldsymbol{y}}d\boldsymbol{y} \tag{60}$$
$$= \eta_\Omega L \left[\hat{K} - \hat{K}L^\dagger \left(\eta_\Omega^{-1} + L\hat{K}L^\dagger\right)^{-1} L\hat{K}\right] L^\dagger\phi$$
$$= \eta_\Omega \left[(L\hat{K}L^\dagger) - (L\hat{K}L^\dagger)\left(\eta_\Omega^{-1} + (L\hat{K}L^\dagger)\right)^{-1} (L\hat{K}L^\dagger)\right]\phi$$
$$= \eta_\Omega \left[\eta_\Omega^{-1}\left(\eta_\Omega^{-1} + (L\hat{K}L^\dagger)\right)^{-1} (L\hat{K}L^\dagger)\right]\phi$$
$$= \left(\eta_\Omega^{-1} + (L\hat{K}L^\dagger)\right)^{-1} (L\hat{K}L^\dagger)\phi,$$

where to simplify the notation in places where there is no indication of position the position is $\boldsymbol{x}$. Similarly, the boundary contribution,

$$\eta_{\partial\Omega} L \int_{\partial\Omega} \left[ [\hat{K}^{-1} + \eta_\Omega L^\dagger L]^{-1} \right]_{\boldsymbol{xz}} g(\boldsymbol{z}) d\boldsymbol{z} \tag{61}$$

$$= \eta_{\partial\Omega} L \left[ \hat{K} - \hat{K}L^\dagger \left( \eta_\Omega^{-1} + L\hat{K}L^\dagger \right)^{-1} L\hat{K} \right] g$$

$$= \eta_{\partial\Omega} \left[ (L\hat{K}) - (L\hat{K}L^\dagger) \left( \eta_\Omega^{-1} + (L\hat{K}L^\dagger) \right)^{-1} (L\hat{K}) \right] g$$

$$= \eta_{\partial\Omega} \left[ 1 - (L\hat{K}L^\dagger) \left( \eta_\Omega^{-1} + (L\hat{K}L^\dagger) \right)^{-1} \right] L\hat{K}g$$

$$= \frac{\eta_{\partial\Omega}}{\eta_\Omega} \left( \eta_\Omega^{-1} + (L\hat{K}L^\dagger) \right)^{-1} L\hat{K}g$$

$$= \frac{\eta_{\partial\Omega}}{\eta_\Omega} \int_{\partial\Omega} \left[ \left( \eta_\Omega^{-1} + (L\hat{K}L^\dagger) \right)^{-1} L\hat{K} \right]_{\boldsymbol{xz}} g(\boldsymbol{z}) d\boldsymbol{z}$$

Consequently, $Lf - \phi$ is

$$Lf(\boldsymbol{x}) - \phi = \left[ \left( \eta_\Omega^{-1} + (L\hat{K}L^\dagger) \right)^{-1} (L\hat{K}L^\dagger) - 1 \right] \phi + \frac{\eta_{\partial\Omega}}{\eta_\Omega} \left( \eta_\Omega^{-1} + (L\hat{K}L^\dagger) \right)^{-1} L\hat{K}g \tag{62}$$

$$= -\eta_\Omega^{-1} \left( \eta_\Omega^{-1} + L\hat{K}L^\dagger \right)^{-1} \phi + \frac{\eta_{\partial\Omega}}{\eta_\Omega} \left( \eta_\Omega^{-1} + (L\hat{K}L^\dagger) \right)^{-1} L\hat{K}g$$

we thus find that $\eta_{\partial\Omega} L\hat{K}g$ acts as an additional source term. While it may seem it diverges with $\eta_{\partial\Omega}$ we recall that $\hat{K}$ goes to zero at this limit for arguments on the boundary, hence its contribution is finite and the overall $\eta_\Omega^{-1}$ ensures this quantity $Lf - \phi$ goes to zero. Changing the basis to the eigenfunction basis of $L\hat{K}L^\dagger$ leads to the spectral bias result of the main text.

Last, we note in passing that $\hat{K}$ as the interpretation of the dataset-averaged posterior covariance, given that one introduced only boundary points and fixes them to zero Cohen et al. (2021). Thus, as $n_{\partial\Omega} \to \infty$, $\hat{K}$ involving any boundary point, is zero.

From a practical point of view, one can obtain an estimate for $\hat{K}$ at small, $\eta_{\partial\Omega}$. A straightforward expansion of this quantity to its leading order is then,

$$\hat{K}_{\boldsymbol{xy}} = K(\boldsymbol{x}, \boldsymbol{y}) - \eta_{\partial\Omega} \int_{\partial\Omega} K(\boldsymbol{x}, \boldsymbol{z}) K(\boldsymbol{z}, \boldsymbol{y}) d\boldsymbol{z} \tag{63}$$

$$- \eta_{\partial\Omega}^2 \int_{\partial\Omega} K(\boldsymbol{x}, \boldsymbol{z}) K(\boldsymbol{z}, \boldsymbol{z}') K(\boldsymbol{z}', \boldsymbol{y}) d\boldsymbol{z} d\boldsymbol{z}' + O(\eta_{\partial\Omega}^3)$$

which can then be evaluated analytically in some cases.

