$$= \int e^{-\sum_{\alpha=1}^p \mathcal{S}_{\text{eff}}[f_\alpha]} \prod_\alpha^p \mathcal{Z}_\alpha^{-1} df_\alpha \tag{12}$$

where in the third transition, we assume that if the $i$th point is in the bulk then $\boldsymbol{x}_i \sim \mu_\Omega(\boldsymbol{x})$, otherwise, $\boldsymbol{x}_i \sim \mu_{\partial\Omega}(\boldsymbol{x})$, in addition, all points are chosen to be independent and identically distributed. With a slight abuse of notation, we take the number of data points in the bulk $n_\Omega \sim \text{Pois}(\bar{\eta}_\Omega)$, and on the boundary $n_{\partial\Omega} \sim \text{Pois}(\bar{\eta}_{\partial\Omega})$ as independent Poisson-distributed random variables. We denote by $\bar{\eta} = \bar{\eta}_\Omega + \bar{\eta}_{\partial\Omega}$, where $\bar{\eta}_\Omega$ and $\bar{\eta}_{\partial\Omega}$, the expectation of the Poisson variables on the bulk and boundary, are the true deterministic number of points in the bulk and on the boundary, respectively. The effective action is then defined as:

$$\mathcal{S}_{\text{eff}}[f] = \bar{\eta} - \bar{\eta}_\Omega \mathbb{E}_{\boldsymbol{x}} \left[ e^{-\mathcal{S}_\Omega[f|\boldsymbol{x}]} \right] - \bar{\eta}_{\partial\Omega} \mathbb{E}_{\boldsymbol{x}} \left[ e^{-\mathcal{S}_{\partial\Omega}[f|\boldsymbol{x}]} \right]$$
$$+ \frac{1}{2} \int_\Omega \int_\Omega f(\boldsymbol{x})[K^{-1}](\boldsymbol{x}, \boldsymbol{y}) f(\boldsymbol{y}) d\boldsymbol{y} d\boldsymbol{x}. \quad (13)$$

Next, Taylor expanding the above exponent, we obtain the following effective action in first-order

$$\mathcal{S}_{\text{eff}}[f] = \bar{\eta}_\Omega \mathbb{E}_{\boldsymbol{x}} \left[ \mathcal{S}_\Omega[f|\boldsymbol{x}] \right] - \bar{\eta}_{\partial\Omega} \mathbb{E}_{\boldsymbol{x}} \left[ \mathcal{S}_{\partial\Omega}[f|\boldsymbol{x}] \right]$$
$$+ \frac{1}{2} \int_\Omega \int_\Omega f(\boldsymbol{x})[K^{-1}](\boldsymbol{x}, \boldsymbol{y}) f(\boldsymbol{y}) d\boldsymbol{x} d\boldsymbol{y} + O(\bar{\eta} S_{\text{PDE}}^2), \quad (14)$$

where $\sigma^2 = \sigma_\Omega^2 + \sigma_{\partial\Omega}^2$. Note that, the first-order action decouples the replica modes, which drastically simplifies the analysis . Taking higher-order corrections into account would lead to a tighter prediction. Taking the standard GP limit of infinite sample size proportional to the noise level and using the fact that the square distance between the estimate and the target decreases as well with $n$, these higher-order terms decrease to zero asymptotically in $n$. Figure (1) of the main text