# OpenReview forum: "Spectral-Bias and Kernel-Task Alignment in Physically Informed Neural Networks"
_ICLR.cc/2024/Conference — Submitted to ICLR 2024_

### Official Review · Reviewer_XJUJ · 2023-10-19

**Soundness:** 3 good
**Presentation:** 2 fair
**Contribution:** 3 good
**Rating:** 6
**Confidence:** 2

**Summary:**

This paper derives the so-called neurally-informed equation (NIE) for a PINN, which characterizes the neural network predictions to the solution of a partial differential equation that takes a kernel term reflecting the prior physical knowledge into account. Based on the NIE, the paper proposes a spectral-bias statement that characterizes the training of an infinite-width PINN.

**Strengths:**

* The paper theoretically derives a spectral decomposition statement of the PINN output, which is promising in both understanding the dynamics of the PINN throughout training and designing techniques to improve training.

**Weaknesses:**

* The clarity of the paper needs to be improved. As a theoretical paper, one cannot tell the story by throwing out equations without explaining its assumptions. The two main results (i.e., NIE and spectral bias) need to be formalized into theorems.
* The paper needs more discussion about why the main results are important. In `section 2.1` and `section 2.3`, a lot of focus has been put on discussing how the main results are derived. However, an intuitive explanation of why they are useful is missing.
* As far as I understand, the spectral bias in this paper is based on the assumption that the PINN is infinitely wide. In practice, of course, every NN has a finite width. It is not clear how much things will break down when we look at a finite-width NN.
* Since the main text is 7-page right now, I recommend the authors add more introduction to the background of this work. The latter part of `section 1` goes a bit too fast and cannot be easily comprehended without a solid familiarity with theory of PINNs.

**Questions:**

* In your spectral bias statement, can you make any general statements about what the eigenvalues and eigenvectors will look like, or is it a very case-specific thing?

---

> ### Author Response · Authors · 2023-11-20
> **Authors' reply part I**
>
> _The paper theoretically derives a spectral decomposition statement of the PINN output, which is promising in both understanding the dynamics of the PINN throughout training and designing techniques to improve training.
> Weaknesses:
> The clarity of the paper needs to be improved. As a theoretical paper, one cannot tell the story by throwing out equations without explaining its assumptions. The two main results (i.e., NIE and spectral bias) need to be formalized into theorems._
>
> **Response**: We thank the referee for those comments. We agree that too many derivation details were carried through to the appendix. As a result, we now added a new derivation outline section (section 2 in the revised version) which goes through the main approximations involved in obtaining this result for both linear and non-linear differential operators.
>
> _The paper needs more discussion about why the main results are important. In section 2.1 and section 2.3, a lot of focus has been put on discussing how the main results are derived. However, an intuitive explanation of why they are useful is missing._
>
> **Response**: In this work, we derive theoretical results that give the dataset-average prediction of a PINN. It is a fundamental building block of what we hope would be a fruitful theoretical investigation into spectral bias. Similar works in the context of supervised learning (https://www.nature.com/articles/s41467-021-23103-1) have received much and justified attention.
>
> That being said, by quantifying spectral bias, we can specify what PDE-solutions are best aligned with the DNN's architecture. This is the context of Fig. 2. showing that (a) performance correlates with how well the target lays in the dominant eigenmodes of the kernel. As shown there, in panel (a) when the cumulative spectral distribution function increases slowly, meaning that the target is spread over many kernel eigenmodes--- performance deteriorates. In panel (b) we demonstrate that more data allows for the learning of lower kernel eigenmodes as reflected in the cumulative spectral function of the learned function.
>
> A discussion in this spirit has been added to Sec. 4.
>
> _As far as I understand, the spectral bias in this paper is based on the assumption that the PINN is infinitely wide. In practice, of course, every NN has a finite width. It is not clear how much things will break down when we look at a finite-width NN._
>
> **Response**: The referee is asking a very good question which has been studied in depth in the context of supervised learning (see for instance https://proceedings.neurips.cc/paper/2020/file/ad086f59924fffe0773f8d0ca22ea712-Paper.pdf). For fully connected networks, such as those commonly used in PINN, the performance of infinite-width GP limits is often on par or better than the actual finite neural network. This seems to be related to the lack of feature learning in such networks. Turning to CNN, finite networks do excel over their infinite-width counterparts, but GPs still seem to be a good approximation. In particular, the benefit of clever architecture choices is reflected in the infinite-width GP limit (see for instance https://arxiv.org/abs/1810.05148 for the beneficial effect of pooling on both CNN and their GP limits)
>
> _Since the main text is 7-page right now, I recommend the authors add more introduction to the background of this work. The latter part of section 1 goes a bit too fast and cannot be easily comprehended without a solid familiarity with theory of PINNs._
>
> **Response**: We thank the referee for pointing out these issues. The current version now contains far more introductory material. In particular, a detailed explanation of the PINN setting and the training dynamics we use, an introduction to our use of replicas, and dataset averaging techniques. Further demonstrations of the NIE for non-linear operators and the variational approach we take.

---

> ### Author Response · Authors · 2023-11-20
> **Authors' reply part II**
>
> _Questions:
> In your spectral bias statement, can you make any general statements about what the eigenvalues and eigenvectors will look like, or is it a very case-specific thing?_
>
> **Response**: The referee is asking a very good question. On the one hand, spectral bias reflects both the DNN's architecture and the dataset measure. In supervised learning, architecture and datasets play a major role in DNN performance. We thus expect spectral bias to be a rather rich quantity that adapts to the data and architecture in intricate ways. In these circumstances, one must resort to numerical method to detect and interpret spectral bias (specifically diagonalizing the kernel on a large and finite dataset, finding an ansatz based on a linear combination of outputs-per-neuron of the DNN which matches the eigenvector on the finite dataset but allows extrapolating to arbitrary points).
>
> Luckily PINNs are much simpler in this context. First of all, dataset measures, at least in vanilla settings, are uniform and low dimensional. Hence, one can directly visualize the eigenfunctions of the kernel (unlike in say CIFAR-10 where these would be non-linear functions in dimension 3072) and focus on the architecture's effect on spectral bias. For FCNs and data domains (i.e. rectangular boundaries or no boundaries), we generally expect these modes to correspond roughly to Fourier modes and the difference between architectures would lay mainly in their different spectral decay. So, for instance, trying to use it for equations with multiscale phenomena such as in turbulence,  one would need to design FCN with a much slower spectral decay than in our FCN experiments. This can potentially be done by taking very deep FCNs and tuning these to the "edge of criticality". Still, an experiential demonstration of this is beyond the scope of the current work.

---

### Official Review · Reviewer_bD4a · 2023-10-30

**Soundness:** 2 fair
**Presentation:** 2 fair
**Contribution:** 2 fair
**Rating:** 3
**Confidence:** 4

**Summary:**

This paper Leverages an equivalence between infinitely over-parameterized neural networks and Gaussian process regression (GPR) to derive the neurally-informed equation that governs PINN prediction in the large data-set limit. This equation augments the original one by a kernel term reflecting architecture choices. Spectral analysis is also performed.

**Strengths:**

Derive an alternative form for loss functions for PINN.
Use Several examples to demonstrate the accuracy of equivalent loss.
Perform spectral analysis.

**Weaknesses:**

1 experiments are too simple.
2 lack of reference, such as
Neural-network-based approximations for solving partial differential equations. communications in Numerical Methods in Engineering, 1994
Training behavior of deep neural network in frequency domain.
On the spectral bias of deep neural networks.
The convergence rate of neural networks for learned functions of different frequencies.
A fine-grained spectral perspective on neural networks.
On the exact computation of linear frequency principle dynamics and its generalization.
3 writing can be improved, such as deepdxe-->deepxde
4 I cannot see a clear advantage of NIE form

**Questions:**

why experiments are only 1d simple equations?
what is the advantage of the proposed form?
Fig. 2 is hard to be understood. I also do not know how to use spectral analysis to understand the different results.

---

> ### Author Response · Authors · 2023-11-20
> **Authors' reply**
>
> _Strengths: Derive an alternative form for loss functions for PINN. Use Several examples to demonstrate the accuracy of equivalent loss..._
>
> **Response**: We thank the referee for pointing out strengthening points. However, we find this phrasing together with a similar one below quite odd as we do not derive an alternative loss function for PINNs. We derive an equation whose solution is equivalent to running the PINN on infinitely many datasets and averaging the resulting solution over those.
>
> _Weaknesses:1 experiments are too simple._
>
> **Response**: Perhaps this is related to the issue with the loss function. Our theory paper derives, for the first time (as far as we know), the expected generalization error of a PINN network averaged over all possible dataset draws. Similar theoretical works, in the realm of supervised learning, have received much attention [e.g. https://www.nature.com/articles/s41467-021-23103-1]. We thus strongly feel that a theoretical gap has been filled here. The experiments we carry out are mainly meant to demonstrate the theory at its most basic level.
>
> _2 lack of reference, such as Neural-network-based approximations for solving partial differential equations. communications in Numerical Methods in Engineering, 1994 Training behavior of deep neural network in frequency domain. On the spectral bias of deep neural networks. The convergence rate of neural networks for learned functions of different frequencies. A fine-grained spectral perspective on neural networks. On the exact computation of linear frequency principle dynamics and its generalization. 3 writing can be improved, such as deepdxe-->deepxde 4 I cannot see a clear advantage of NIE form_
>
> **Response**: We thank the referee for pointing out these references. Due to lack of time and the desire to reference these properly, we will add those to the next version.
>
> _Questions: why experiments are only 1d simple equations?_
>
> **Response**: As mentioned before, this is a theory paper and not a new PINN training scheme. the experiments are meant to demonstrate basic applications of the theoretical tool we developed. We also comment that one of our experiments is on a 2D equation (space plus time).
>
> _what is the advantage of the proposed form?_
>
> **Response**: It is not clear to us what is the comparison here. An advantage over what? currently, there is no similar analytical equation governing the PINN data-averaged output. That being said, several parts of the manuscript including Fig. 2. are dedicated to showing how the spectral bias encapsulated in this equation affects learning.
>
> _Fig. 2 is hard to be understood. I also do not know how to use spectral analysis to understand the different results._
>
> **Response**: At the basic level, the source term in the equation should be spanned by high eigenfunctions of the $L \hat{K} L^{\dagger}$ kernel operator. In various simple cases (e.g. linear equation spatially homogenous apart from a source term, and set in a simple domain), these eigenfunctions are roughly Fourier modes and the statement simplifies to saying that simple PINNs work better when the source has only low-frequency modes. Even then, there is the question of how fast do the eigenvalues associated with higher Fourier modes decay. An exponential decay with wave number, as we observe for simple networks, would make it very difficult to capture multiscale phenomena such as turbulence or shock waves.
>
> Next considering richer domains, inhomogeneous operators, etc... our results guide the practitioner to what is the right analog of high Fourier modes. The practitioner can then try and ``spectrally engineer" the kernel such that the source term has larger support on the (analogous-to) high Fourier modes. Such support is captured by our cumulative spectral function graphs ($A_k$) where a faster-increasing function means better support and better expected performance.
>
> Figure 2 is a demonstration of this spectral dependence of learning. The function being learned by the PINN is predominately made of high spectral kernel eigenvalues and lower modes are slowly learned as more and more datapoints are given.
>
> While the rough notion of spectral bias is expected in PINN, our work places it on a formal and quantitative theoretical footing.
>
> We note by passing (and can provide an appendix on this) that one can readily visualize these eigenfunctions by: 1. Obtain the kernel between $n$ data points. Diagonalize the matrix to obtain eigenvectors. Find a linear combination of network's outputs per neuron which coincides with the former eigenvector on the discrete set of datapoints. Note that, provided $n$ is larger than the width of the network, this is a well-posed problem. Use the resulting linear combination to approximation the kernel eigenfunction. Plot it on the domain.
>
> Following the various additions to the text, we hope the referee will give our paper a second read.

---

### Official Review · Reviewer_qyxK · 2023-10-31

**Soundness:** 2 fair
**Presentation:** 2 fair
**Contribution:** 3 good
**Rating:** 6
**Confidence:** 3

**Summary:**

The paper considers the task of solving a given PDE via training a physically informed neural network (PINN) and aims at a theoretical description of the solution obtained in this way. To this end, they consider infinitely wide PINN trained until convergence by gradient descent with white noise added to the gradients. Under replica trick and first-order Taylor expansion, the authors show that PINN solution is described by linear integral equation - Neural informed equation (NIE). Then, the authors proceed with revealing a spectral picture behind NIE by identifying another operator whose eigendecomposition decouples PINN learning in independent spectral components. The authors demonstrate their formalism theoretically on a toy example of a simple first-order differential equation on a positive real line, as well as numerically for 1D heat equation with a specific source term.

**Strengths:**

The main results of the paper - NIE equation (5) and its spectral decomposition (15,16) look like a solid development that can become a basis for future analysis of the PINN behavior. Importantly, the approximations done when deriving NIE may indeed be negligible, as demonstrated by a good agreement between direct simulation and NIE solution for big dataset sizes in Figure 1.

My score would be significantly higher if not for the presentation issues described below.

**Weaknesses:**

The main weakness of the paper is not clear enough presentation, making the accurate reading of the paper (e.g. by trying to verify theoretical results) quite difficult. Given the almost 2 unspent pages of the main paper and quite the small size of the current appendix, there is a lot of room for giving the necessary details.

**The setting** is not described clearly and in full detail. For example, the authors mention that posterior (3,4) describes infinitely wide NN trained by Gradient Descent with white noise the gradients to the gradients weight decay. Apparently, this setting follows [Naveh2021], but this is not mentioned explicitly by the authors, and the respective reference is mentioned together with classical NTK work [Jacot2018]. Since the setting with added white noise to the gradient is much less common compared to that of [Jacot2018], it may take a significant time to figure out what the authors actually mean. Moreover, as mentioned by [Naveh2021], the considered setting is equivalent to Bayesian training with NNGP kernel, adding more options for confusion (see, for example, these two references at the end of page 3).

**Derivations** often lack comments and/or intermediate steps required for their understanding. A few examples with associated questions are
1. While it is expected that the result of [Naveh2021] for standard fully-connected networks carries over to the physically-informed networks and resulting in the posterior in eq. (3,4), the clarity of presentation would be improved if the authors would give the reference for the respective statement (if it was obtained previously) or formulate and prove a proposition (even if it turns out to be quite simple).
2. Derivation of eq. (5) uses the replica trick to evaluate the average of possible choices of the training dataset $\mathcal{D}_n$. I would expect that the replica trick is a technique that a typical member of ICLR community does not known by default, so at least a reference or a short description of the technique is needed to make derivation accessible to a broader audience.
3. Is the last transition in (31) correct? Somehow, it takes sum over replica modes out of the inner exponent in the action, thus decoupling the modes. It seems that decoupling is only possible after Taylor expands this exponent up to the first order (as also mentioned by the authors after eq. (33)).
4. After decoupling replica modes, the authors continue to work with the effective action (33) of a single mode. This raises a number of questions: (i) whether the replica limit $p\to0$ was taken implicitly at some point? (ii) why does this effective action describe the (data averaged) posterior distribution of $f$ we are interested in?
5. The authors first state that they consider uniform data distribution in the bulk $\Omega$, but in the toy example, consider $\Omega=\mathbb{R}_+$ with $|\Omega|=\infty$ making uniform distribution on $\Omega$ not well defined. How is this issue resolved?
6. Calculation behind eq. (9) are not trivial and involve additional assumptions, e.g. vanishing of the boundary $x=\infty$ term during integration by parts.
7. Contour integration performed to obtain Green function $G(x,x')$ involves an approximation of taking into account only a single pole. This is mentioned in a few lines of text without even specifying which of the poles dominates the result. More detailed justification is required for this approximation to be acceptable (e.g. providing numerical evidence of the single pole dominating the integral).
8. I could not understand how NIE equation (5) can be rewritten in the form (46) without integration by parts. Could you please provide more details?

As can be seen from the above, there are a number of assumptions and approximations performed in the paper. More careful commenting and/or validation of these assumptions would make the claims of the paper more verifiable. A separate example of that is the statement that the proposed approach can be extended to the nonlinear operators $L$. However, this claim is supported only by a single line at the end of Appendix B.2, which hints at a potential derivation strategy for nonlinear operators $L$. If not described in more detail (e.g. by providing execution of the derivation strategy suggested by the authors), this claim seems suitable only for the discussion section (as an example of a direction for future work).

Regarding the plots, there is a small issue of visibility of axes and colormap ticks values on the inset heatmap plots from Figure 2, which are required to support the benefit of stronger concentration of cumulative spectral functions. At the moment, the only way to see them is with extremely high zoom and good eyesight.

A few (potential) minor typos:
- Tilde is missing in the right-hand side of the reparametrization of variation from $h$ to $\widetilde{h}$ (eq. after eq. (35)).
- Is there an extra index $k$ in the definition of cumulative spectral function (e.g. it should be $A_k[Q,f]=\|P_k f\|^2 / \|f\|^2$)?

[Naveh2021] Gadi Naveh, Oded Ben David, Haim Sompolinsky, and Zohar Ringel. Predicting the outputs of finite deep neural networks trained with noisy gradients.

[Jacot2018] Arthur Jacot, Franck Gabriel, and Clément Hongler. Neural tangent kernel: Convergence and generalization in neural networks.

**Questions:**

Please refer to the above section.

---

> ### Author Response · Authors · 2023-11-20
> **Author's reply part I**
>
> We'd like to thank the referee for his/her diligent and careful work reviewing this manuscript.
>
> _The main weakness of the paper is not clear enough presentation........_
>
> **Response**: We agree with the referee that too many derivation details were carried through to the appendix. As a result, we now added a new derivation outline section (section 2 in the revised version) which goes through the main approximations involved in obtaining this result for both linear and non-linear differential operators.
>
> _The setting is not described clearly and in full detail..._
>
> **Response**: Following the referee's advice, we now clearly state what training protocol we follow and how it matches Bayesian inference with Gaussian-noisy observation. We comment that this training procedure is close in spirit to NTK. Both approaches take infinite width, vanishing learning rates, and MSE loss. Our version adds noise to the gradients and weight decay to make closer contact with Bayesian sampling as $t \rightarrow \infty$. For any finite $t$, taking the noise to zero, the two training protocols coincide. In practice, our Bayesian posterior, also referred to sometimes as Bayesian posterior with the NNGP kernel, is on par with NTK in terms of performance as shown in various works [see for instance https://proceedings.neurips.cc/paper/2020/hash/ad086f59924fffe0773f8d0ca22ea712-Abstract.html].
>
> _Derivation issue 1._
>
> **Response**: The revised setting section now hopefully clarifies this point better. See specifically Eq. (3).
>
> _ Derivation issue 2._
>
> **Response**: The new "outline of derivation" section (Sec. 2. 2nd paragraph) now hopefully discusses this more clearly.
>
> _Derivation issue 3._
>
> **Response**: We thank the referee for her/his diligent work in spotting this issue. We erroneously made reference to an ill-defined quantity (which we never really use) at the end of this equation. This last equality has now been removed with no consequences for the rest of the work.
>
> _ Derivation issue 4._
>
> **Response**: The referee is right to point out this step which the text unfortunately glanced over. The revised version contains an extra equation explaining this (see discussion below equation 37). Specifically, once replicas are decoupled, the data-averaged replicated partition function ($\langle Z^p \rangle_{data}$) takes the form, $\langle Z \rangle_{data}^p$ hence yielding $\log(Z) =_{\lim p \righarrow 0} (1-\lange Z^p \rangle_{data})/p=\log(\langle Z\rangle_{data})$. Thus we can use the single replica partition as the posterior partition function.
>
> _Derivation issue 5.+6._
>
> **Response**: Again, the referee is spot on. The formal way of phrasing what we do is we cutoff the semi-infinite interval at $L$ and define $\eta$ such that a uniform *density* of points is maintained as we take $L$ to infinity. The resulting action from this procedure is the one we used. Our results do not depend on $L$, in the large $L$ limit, since the solution of the equation decays exponentially on a scale related to the density. Our numerical experiments are of course carried on a finite $L=512$ (much greater than the scale of kernel decay) with the number of collocation point scaling as $\Lambda$ times this density. This is now clarified in the appendix. We will provide more rigorous arguments for the expected decay of boundary terms (allowing us to ignore $1/L$ effects), in the next version of the paper.

---

> ### Author Response · Authors · 2023-11-20
> **Authors reply part II**
>
> _Derivation issue 7._
>
> **Response**: The single pole approximation is indeed a rather uncontrolled one at the moment. In particular, it gives a finite derivative around zero (via $|x|$) whereas the exact Green's function should be smooth around this point. We did find that it captures the rough envelope of decay quite Ok, and hence gives a useful physical picture. The current and previous versions openly stated that the single pole's accuracy was just a numerical observation.
>
> To avoid these discrepancies, in the numerics involved we did not use this approximation but in fact, performed numerical integration to obtain this Green's function (as we hope was also clear in the previous version). We intend to test the effect of extra poles in the next version.
>
> Towards the next version, we will try and improve this approximation, either by giving a more throughout numerical assessment of its accuracy or by providing better analytical guarantees. Some analytical observations towards this point are that (i) Taking $l=\eta=1$ for simplicity, the simple poles obey $e^{k_p^2}+k^2_p=0$ hence for $k = k_p + \delta k$, $e^{k^2}+k^2 = (2k_p e^{k_p^2}+2k_p)\delta k=3(k_p - k_p^3) \delta k$. This together with the $e^{ikx}$ Fourier factor this yields a residue going like $e^{i k_p x}/(k_p - k_p^3)$. (ii) At large $|k_p|$ poles can appear only near lines in the complex plane which lay along $(\pm 1 + i)$, as these are the only regions where $|e^{k^2}|$ is not exponentially decaying or exploding. As a result poles with large $k_p$, have a sizable imaginary part and hence have exponentially decaying residue. (iii) $k_p$'s are related to the $p$ branch of Lambert W function at $z=1$. Provided these do not have some bunching property (which seems unlikely) then following points (i) and (ii) that the contribution of poles with large $p$ becomes negligible.
>
>
> _Derivation issue 8._
>
> **Response**: This is due to our $L^{\dagger}$ notation, where we state that it means $L$ is acting to the left ($fL^{\dagger}...=(Lf)...$. Thus, we haven't really carried the derivative to act on the right side and formally avoided the integration by parts. For instance if $L=\partial_x$ then our notation means $\int_0^1 dx f L^{\dagger} g \equiv \int_0^1 dx L[f] g$, then following an integration by parts it is equal to $-\int_0^1 dx F L[g]+ fg|_0^1$
>
> _As can be seen from the above, there are a number of assumptions and approximations performed in the paper. More careful commenting and/or validation of these assumptions would make the claims of the paper more verifiable..._
>
> **Response**: The revised manuscript now contains a simple yet typical example of how we can tackle non-linear operators (App. Sec. B3). The new outline-of-derivation section should also provide more clarity on this matter.
>
> _Regarding the plots, there is a small issue of visibility ..._
>
> **Response**: Would this manuscript be accepted, we promise to address these graphical comments which we cannot do at the moment due to technical constraints. We are very thankful to the referee for his/her important insights and careful reading of the text.
>
> _ A few (potential) minor typos..._
>
> ** Response**: Many thanks again. Both have been corrected.

---

### Author Response · Authors · 2023-11-23

We would like to thank all the referees. For your convenience, we have added a version with our main changes highlighted in blue.

---

### Meta-Review · Area_Chair_xN52 · 2023-12-18

**Metareview:**

The paper provides analysis of physically informed neural networks in the infinite width limit with Gaussian noise in the gradient steps - the analysis follows a kernel regime approximation similar to Neural Tangent Kernel (NTK) analysis. The reviewers point out many issues with initial manuscript including lack of sufficient details about the setting, relation to related work, and simplicity of the experiments. The authors fix some issues on clarity on the paper's analysis setting. The meta reviewer also acknowledges that some points in Reviewer bD4a's comments are sloppy and incomplete. Nevertheless, upon reviewing the discussion, the metareviewer believes the concerns around significance of empirical results, and comparisons to related work remain valid. The meta reviewer additionally believes that given the huge literature on kernel regime analysis of neural networks in the past 5+ years, a purely theoretical analysis alone without conclusive empirical demonstration of insights from the derivations is insufficient at this point to have significant impact in the community.

**Justification For Why Not Higher Score:**

Although two reviewers have marginal positive review, and the third reviewer's comments I would consider are somewhat sloppy, I have explained my concerns around yet another ntk style analysis of neural networks not being impactful in the meta review. That said if the SAC/PC feel the contributions are significant, I am ok changing the response.

**Justification For Why Not Lower Score:**

NA

---

### Decision · Program_Chairs · 2024-01-16

Reject